# Brain somatic mutations observed in Alzheimer's disease associated with aging and dysregulation of tau phosphorylation

Jun Sung Park [1,9], Junehawk Lee [2,9], Eun Sun Jung[3,4], Myeong-Heui Kim [1], Il Bin Kim[5], Hyeonju Son[6], Sangwoo Kim [6], Sanghyeon Kim[7], Young Mok Park[8], Inhee Mook-Jung[3,4], Seok Jong Yu [2] & Jeong Ho Lee[1,5]

The role of brain somatic mutations in Alzheimer's disease (AD) is not well understood. Here, we perform deep whole-exome sequencing (average read depth 584×) in 111 postmortem hippocampal formation and matched blood samples from 52 patients with AD and 11 individuals not affected by AD. The number of somatic single nucleotide variations (SNVs) in AD brain specimens increases significantly with aging, and the rate of mutation accumulation in the brain is 4.8-fold slower than that in AD blood. The putatively pathogenic brain somatic mutations identified in 26.9% (14 of 52) of AD individuals are enriched in PI3K-AKT, MAPK, and AMPK pathway genes known to contribute to hyperphosphorylation of tau. We show that a pathogenic brain somatic mutation in *PIN1* leads to a loss-of-function mutation. In vitro mimicking of haploinsufficiency of PIN1 aberrantly increases tau phosphorylation and aggregation. This study provides new insights into the genetic architecture underlying the pathogenesis of AD.

[1] Biomedical Science and Engineering Interdisciplinary Program, Korea Advanced Institute of Science and Technology (KAIST), Daejeon 34141, Republic of Korea. [2] Center for Supercomputing Applications, Division of National Supercomputing, Korea Institute of Science and Technology Information, Daejeon 34141, Republic of Korea. [3] Department of Biochemistry and Biomedical Sciences, College of Medicine, Seoul National University, Seoul 03080, Republic of Korea. [4] Neuroscience Research Institute, College of Medicine, Seoul National University, Seoul 03080, Republic of Korea. [5] Graduate School of Medical Science and Engineering, Korea Advanced Institute of Science and Technology (KAIST), Daejeon 34141, Republic of Korea. [6] Department of Biomedical Systems Informatics, Brain Korea 21 PLUS Project for Medical Science, Yonsei University College of Medicine, Seoul 03722, South Korea. [7] Laboratory of Brain Research, Stanley Medical Research Institute (SMRI), 9800 Medical Center Drive, Suite C-050, Rockville, MD 20850, USA. [8] Center for Cognition and Sociality, Institute for Basic Science (IBS), Daejeon 34126, Republic of Korea. [9] These authors contributed equally: Jun Sung Park, Junehawk Lee. Correspondence and requests for materials should be addressed to S.J.Y. (email: codegen@kisti.re.kr) or to J.H.L. (email: jhlee4246@kaist.ac.kr)

Somatic mutations are post-zygotic genetic variations that are not inherited from one's parents and result in genetically different cells within a single organism[1]. In the brain, somatic mutations are known to arise and accumulate during development and with increasing age, possibly due to DNA replication errors and extensive oxidative stress followed by gradual defects in DNA repair mechanisms[2,3]. Recently, we and other groups have demonstrated that brain somatic mutations arising from the ventricular or subventricular zone, a well-known neural stem cell niche in the human brain, lead to various childhood or adult neurological disorders, including cortical malformations, intractable epilepsies, and brain tumor[4,5]. However, the significance or the pathogenic roles of brain somatic mutations in neurodegenerative disorders such as Alzheimer's disease (AD) remain unclear.

AD is a devastating neurodegenerative disorder and the most predominant form of dementia. It affects about 10% of older adults aged 65 years and older[6]. AD is neuropathologically characterized by the presence of extracellular β-amyloid plaques and intracellular neurofibrillary tangles composed of hyperphosphorylated tau protein[7]. Century-long investigations have sought to identify the molecular genetic causes of AD and have documented autosomal dominant pathogenic germline mutations in APP[8], PSEN1[9], and PSEN2[10] in early-onset familial AD and disease-associated single-nucleotide polymorphisms (SNPs) in APOE[11], TREM2[12], and others in late-onset sporadic AD. These germline mutations, however, only account for, at most, 50% of all sporadic AD cases;[13,14] the genetic etiology of the other half of sporadic AD cases remains unclear.

Growing evidence suggests that neurofibrillary tangles in the entorhinal cortex and hippocampus in the early stages of AD can spread to other brain regions and act as local initiators of further aggregation thereof in a prion-like fashion[15,16]. Meanwhile, the progression of neurofibrillary tangle pathology throughout the brain has been shown to be strongly correlated with the severity of cognitive impairment in AD patients[17]. Given a focal onset of the pathology and progressive spread of protein aggregates in AD[18], one could plausibly speculate that brain somatic mutations in the hippocampal formation (HIF), including the entorhinal and other hippocampal regions, might initially trigger tau pathology in AD. However, the contribution of brain somatic mutations to the initial appearance of tau pathology in the HIF of AD brains is unknown. Here, by performing deep whole-exome sequencing of postmortem HIF and matched blood tissues from 52 AD patients and 11 individuals not affected by AD, followed by functional studies of identified mutations, we provide the direct evidence of the contribution of brain somatic mutations to tau pathology and AD pathogenesis.

## Results

**Bioinformatic analysis pipeline for somatic mutation.** Since somatic mutations accumulate at relatively lower levels in non-cancer samples than in tumors, we sought to initially enrich cells-of-interest to ensure enough supporting reads of altered alleles, thereby allowing us to accurately detect low-level somatic mutations (variant allelic frequency [VAF] <5%) in bulk tissue[19,20]. To do this, we utilized laser capture microdissection (LCM) to isolate and enrich neuronal cells from the entorhinal cortex, subiculum, CA1-4, and dentate gyrus regions of the HIF in frozen brain tissue blocks from 52 AD and 11 individuals not affected by AD (Table 1, Supplementary Fig. 1, and Supplementary Movie 1). We then performed high-depth exome sequencing of enriched HIF tissues and matched blood samples (average sequencing read depths of 564.9× in brain and 598.9× in blood) (Supplementary Data 1). To detect low-level somatic single-nucleotide variations

| Table 1 Summary of baseline characteristics of enrolled subjects |  |  |
|---|---|---|
| **Category** | **AD** | **non-AD** |
| No. of individuals | 52 | 11 |
| Gender | 16 M; 36 F | 7 M; 4 F |
| Age, mean ± SD (range) | 83.06 ± 7.96 (70–96) | 74.09 ± 9.43 (57–89) |
| Onset age, mean ± SD (range) | 73.18 ± 11.67 (48–94) | – |
| Braak tau, mean ± SD (range) | 4.65 ± 1.31 | 0.45 ± 0.52 |
| #High (4–6) | 85% (44/52) | 0% (0/11) |
| #Low (0–3) | 15% (8/52) | 100% (11/11) |
| Family history[a] | 23% (12/52) | 9% (1/11) |
| APOE ε4 genotype (%) |  |  |
| ε4/ε4 | 15.4% (8/52) | 9.1% (1/11) |
| ε4/− | 48.1% (25/52) | 18.2% (2/11) |
| −/− | 36.5% (19/52) | 72.7% (8/11) |

AD Alzheimer's disease, non-AD control unaffected by AD, APOE apolipoprotein E
[a]Positive family history indicates subjects having at least one first- or second-degree relative diagnosed with AD

(SNVs) from high-depth sequencing data, we utilized the sensitive somatic mutation caller MuTect with modified parameter options[21]. Quantitative and qualitative post-filters were subsequently applied on raw calls to rule out false positives. In the post-filtering process, candidate variants were scored using an empirical Bayesian framework (Empirical Bayesian score (EBscore)) to rule out error-prone sites and sequencing errors[22]. An optimal cut-off value for EBscore that maximized the sum of sensitivity and specificity was determined using deep whole-exome sequencing data from an independent cohort of 21 healthy control brains (as a panel of normals) and 11 schizophrenia brains (as a test dataset) and using validation results through targeted amplicon sequencing (Supplementary Fig. 2, Supplementary Data 2, 3, and Methods). EBscore (cut-off of >2.396) and other filtering parameters (e.g., read-depth, variant allele fraction, position of supporting reads, and visual inspection) were applied on raw calls for 63 HIF and 48 blood samples from AD individuals and individuals not affected by AD (Fig. 1a). To validate the performance of the mutation calling pipeline, we randomly picked around ~11% (84/777 SNVs) of filtered somatic SNVs from the HIF specimens and performed targeted amplicon sequencing at read-depths of 5,434× to 4,797,498× to validate the filtered somatic SNVs (Supplementary Data 4). In result, we could ensure that 79.8% of the filtered SNVs were putative somatic SNVs and that true calls showed strong correlations when comparing VAFs across individual platforms (Fig. 1b, c and Supplementary Fig. 3). The VAFs of validated somatic mutations ranged from 0.52% to 15.3% (Supplementary Data 4). Putative blood somatic SNVs were called using the same pipeline. For cases in which only HIF samples were available (15 cases), we processed sequencing data using the same caller and post-filter criteria; however, we applied more strict quantitative filtering options to rule out accidental germline mutations (see Methods). There was no significant difference in the average number or mean VAF of post-filtered brain somatic mutations between AD and non-AD groups with or without peripheral tissues (Supplementary Fig. 4c, d). Overall, we found 760 and 2846 putative somatic SNVs in 63 HIF and 48 blood samples, respectively (Supplementary Data 5).

**Quantitative comparison of somatic SNVs in AD and non-AD.** Next, we examined mutation count, variant allele fraction, and mutation subtypes of brain and blood somatic SNVs in Alzheimer's disease and age-matched unaffected control specimens

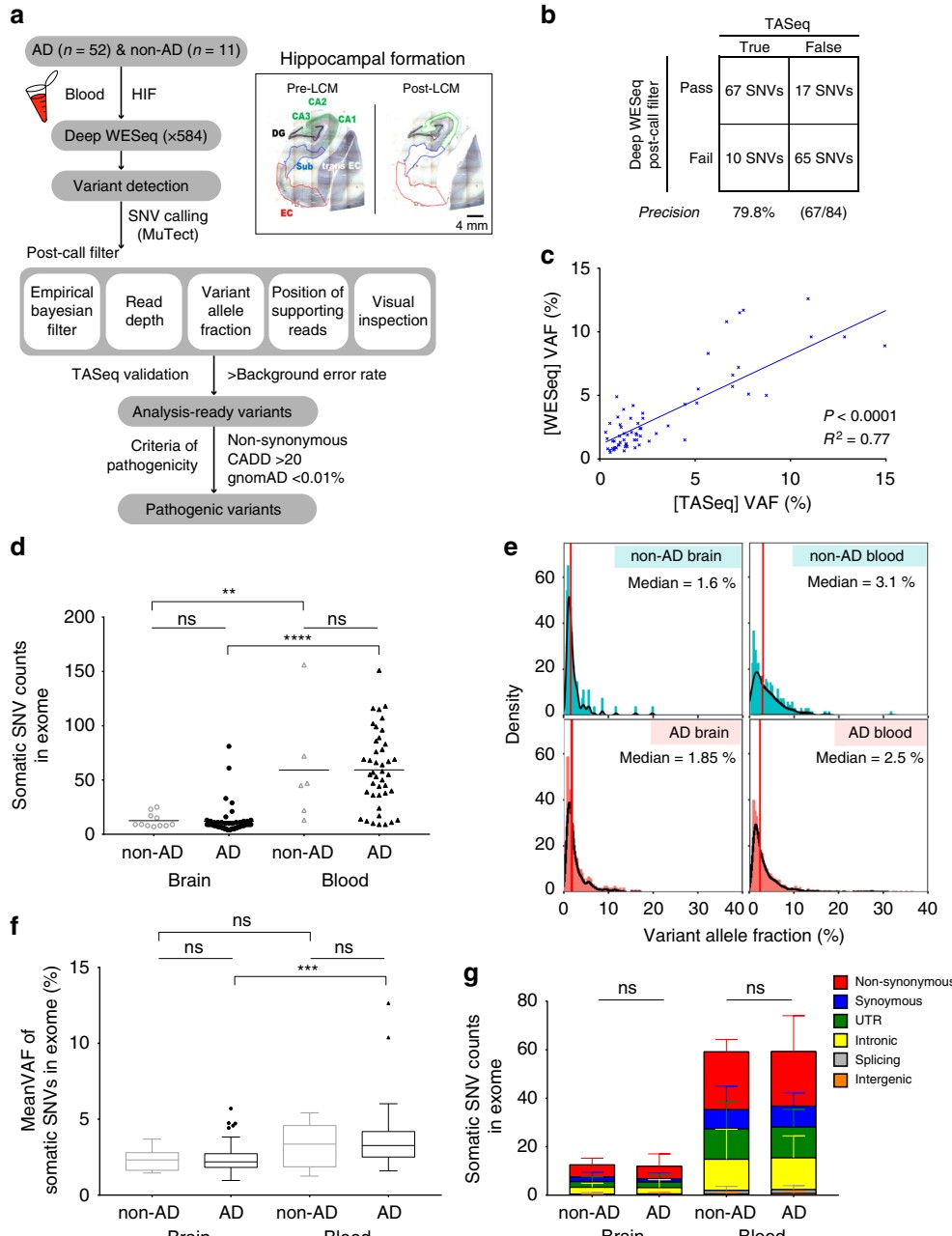

**Fig. 1** Bioinformatic analysis pipeline and quantitative comparison of somatic single-nucleotide variations (SNVs) found in hippocampal formation (HIF) and blood tissues from Alzheimer's disease (AD) and non-AD individuals. **a** Schematic figure for the bioinformatic analysis pipeline used in this study. **b** Validation results of post-filtered brain somatic mutations using targeted amplicon sequencing. Post-filter criteria ensured a precision of 79.8% in detecting filtered somatic SNVs. **c** Correlation of variant allelic frequency (VAFs) for 67 true calls between deep whole-exome sequencing and targeted amplicon sequencing. **d** Comparison of average mutation counts of somatic SNVs in HIF and blood tissues from AD and non-AD individuals. **e** Distribution of VAFs of pooled SNVs in each tissue from AD and non-AD individuals. Median VAF is indicated in red bars on each panel. **f** Comparison of mean VAFs in each group, which are shown as box plots (center line, median; box limits, upper and lower quartiles; whiskers, maximum and minimum values; points, outliers). **g** Contribution of six different mutation subtypes found in each group and data are mean ± SD. $P$ values were calculated by one-way analysis of variance (ANOVA) test, followed by post hoc multiple comparison in **d**, **f**. $P$ values were calculated by two-way ANOVA test, followed by post hoc multiple comparison in **g**. ns, not significant; **$p < 0.01$; ***$p < 0.001$; ****$p < 0.0001$. Source data is available as a Source Data file

(AD brain, non-AD brain, AD blood, and non-AD blood). Although there was no significant difference in the average numbers of brain or blood somatic mutations between AD and non-AD groups, we found that the average numbers of somatic SNVs in HIF tissue (AD = 11.96 SNVs, non-AD = 12.55 SNVs) were significantly lower than those in blood (AD = 59.31 SNVs, non-AD = 59.17 SNVs) (Fig. 1d). The number of mutation

subtypes (e.g., Non-synonymous, Synonymous, UTR, Intronic, Splicing, and Intergenic) for HIF and blood somatic SNVs showed no significant differences between AD and non-AD individuals (Fig. 1g). We also found that 86.4% (657/760) and 76.8% (2186/2846) of all HIF and blood somatic SNVs had VAFs <5%. Despite a higher median VAF in blood samples (AD = 2.5%, non-AD = 3.1%), compared to that in HIF samples

(AD = 1.85%, non-AD = 1.6%), there were no statistically significant differences in mean VAFs in each individual (Fig. 1e, f). These findings indicated that neither mutation count nor the genomic location of the somatic mutations were significantly different between AD and non-AD individuals. In light of the significant differences in the mutation burden between the HIF tissues and blood, however, we speculated that somatic mutations in the HIF and blood may be acquired via different mutational processes.

Accordingly, we sought to outline the biological processes underlying the occurrence and accumulation of somatic mutations in HIFs and blood. An increase of somatic mutations with aging has been reported in cancer[23] and, more recently, in early-onset neurodegeneration due to genetic disorders affecting DNA repair[24,25]. Thus, we examined whether somatic mutations in HIF tissue and blood increase in number in AD individuals in an age-dependent manner. We were able to estimate that somatic SNVs with VAF of at least 0.52% in neuronal cells of HIF increase a rate of 0.53 somatic SNVs per exome per year (Supplementary Fig. 10a); a 4.8-fold faster rate (2.55 SNVs/exome per year) was observed for blood (Supplementary Fig. 10b). When we extrapolated this observation on a broader genomic scale (~75 million bps to ~3 billion bps), we calculated that 22 and 106 somatic SNVs would accumulate in HIF and blood tissues, respectively, every year.

**Mutation signatures of AD brain and blood somatic mutations.** Somatic cells from different tissues are exposed to different intrinsic (e.g., DNA polymerase error, impairment in DNA repair mechanisms) and extrinsic (e.g., tobacco smoking, ultraviolet ray) mutagenic sources[26,27]. These mutational sources elicit distinct mutational patterns in terms of base alteration and their associated nucleotide contexts, known as the mutational signature[26]. To characterize tissue-specific mutational processes, we first pooled all putative somatic SNVs available for signature analyses according to tissue type (AD brain = 595 SNVs, AD blood = 2475 SNVs). We then decomposed all possible combinations of mutation signatures using maximum likelihood estimation and identified the best model[28]. Analyses were conducted for 65 single base substitution (SBS) signatures (adjusted to human whole-exome trinucleotide frequencies) from the PCAWG database[29] (Supplementary Data 6). In AD brains, we found that SBS signatures 5, 1, and 18 accounted for 23.6%, 15%, and 22.2% of all somatic mutations, respectively (Fig. 2a and Supplementary Data 7). Meanwhile, in AD blood, SBS signatures 5 and 1 explained 71.0% and 19.8% of all somatic mutations, respectively (Fig. 2b). Signatures 5 and 1 have been reported to be universally present in almost every cancer sample, suggesting that underlying mutational processes operate continuously as part of the normal aging process, albeit at different rates in individual tissue types[29]. SBS signature 5 has recently been found to cause an accumulation of somatic mutations via a universal genomic aging mechanism that is of yet unknown[30]. SBS signature 1 appears to be generated during DNA replication, eliciting spontaneous deamination of 5-methyl cytosine to thymine, and therefore associated with cell proliferation[26]. Interestingly, SBS signature 18, which presumably indicates DNA damage by reactive oxygen species (ROS)[29], has been found to exhibit the second highest major contribution in AD brains. Therefore, these findings suggest that, while age-related or clock-like mutational processes are crucial for the occurrence and accumulation of somatic mutations in both brain and blood tissues of Alzheimer's disease, DNA damage induced by endogenous or exogenous ROS contributes more to the accumulation of somatic mutations in AD brains than in AD blood.

**Brain somatic mutations associated with tau pathology in AD.** We then wondered whether pathogenic brain somatic SNVs found in AD individuals are significantly associated with common biological processes implicated in AD pathogenesis, compared to blood somatic mutations in AD or brain and blood somatic mutations in non-AD individuals. To identify potentially pathogenic mutations, we initially excluded common variants that were unlikely to be deleterious by filtering out variants with a minor allele frequency of ≥0.01% in the general population according to the genome Aggregation Database (gnomAD r2.0.2)[31]. Utilizing the latest version of Phred-scaled CADD (combined annotation-dependent depletion) score (GRCh38-v1.4), a well-established variant pathogenicity scoring system, we prioritized somatic SNVs according to their scores and considered variants with a scaled CADD score of >20 as putatively pathogenic[32]. In result, 65.3% (175/268) of non-synonymous somatic SNVs in AD brains were predicted as rare and putatively pathogenic variants (Supplementary Data 8). To examine which biological process would be associated with these pathogenic somatic SNVs, we performed gene-set enrichment tests using the KEGG (Kyoto Encyclopedia of Genes and Genomes) database and compared the significances of noted associations[33]. Surprisingly, putatively pathogenic somatic mutations in HIFs of AD individuals were significantly enriched for the PI3K-AKT pathway (Top1, adj. P < 0.0001; 15/341 overlap), mitogen-activated protein kinase (MAPK) pathway (Top3, P = 0.0007; 11/255), and AMP-activated protein kinase (AMPK) pathway (Top6, P = 0.003; 7/124) (Fig. 3a). To validate this finding, we applied random permutation tests (10,000 trials) for all putatively pathogenic somatic mutations found in AD and normal samples, and found the P values of the above associations to be significant (PI3K-AKT pathway, P = 0.0019; MAPK pathway, P = 0.0068; AMPK pathway, P = 0.0237). Also, we showed that gene-length bias did not affect the enrichment test result since the independent gene size-adjusted enrichment test still showed significant P values in all aforementioned pathways (Supplementary Fig. 5 and Methods). As these biological pathways can modulate tau kinase or phosphatase activity, we deemed that alterations in these pathways would likely affect the equilibrium in the phosphorylation status of Tau[34–37] and that putatively pathogenic somatic mutations in HIF of AD could be associated with dysregulation of tau phosphorylation.

Thus, with the use of text-mining engines, we sought to pinpoint which genes with putatively pathogenic somatic mutations could be directly linked to the tau pathology of AD. Using the text-mining engine DigSee[38], we discovered 28 genes with somatic mutations that directly affected the phosphorylation of tau protein (Fig. 3b and Supplementary Data 9). Among them, *PIN1* was the most promising candidate for further functional validation, as it is one of the most frequently reported AD-associated genes and has a high pathogenicity score. As a peptidyl-prolyl *cis–trans* isomerase, PIN1 balances *cis/trans* conformation of phospho-tau[39], and complete loss of its expression in mice reportedly triggers age-dependent hyperphosphorylation of tau and neurofibrillary tangles in murine brains[40]. From the HIF of patient AD-1444, we found a highly pathogenic (CADD = 26.7, 52th pathogenic SNV in AD brain) and novel somatic SNV in *PIN1* (WESeq VAF = 1.8%, c.477C>T, p. Thr152Met) (Fig. 3c). Testing the individual sub-regions of the HIF from this AD individual, followed by targeted amplicon sequencing on the extracted gDNA, we found that the missense variant was universally present in all of the examined sub-regions, with VAFs ranging from 0.51% to 1.62% (Supplementary Data 10). Then, to test whether neurofibrillary tangle-positive neurons specifically carry the mutation in the entorhinal cortex,

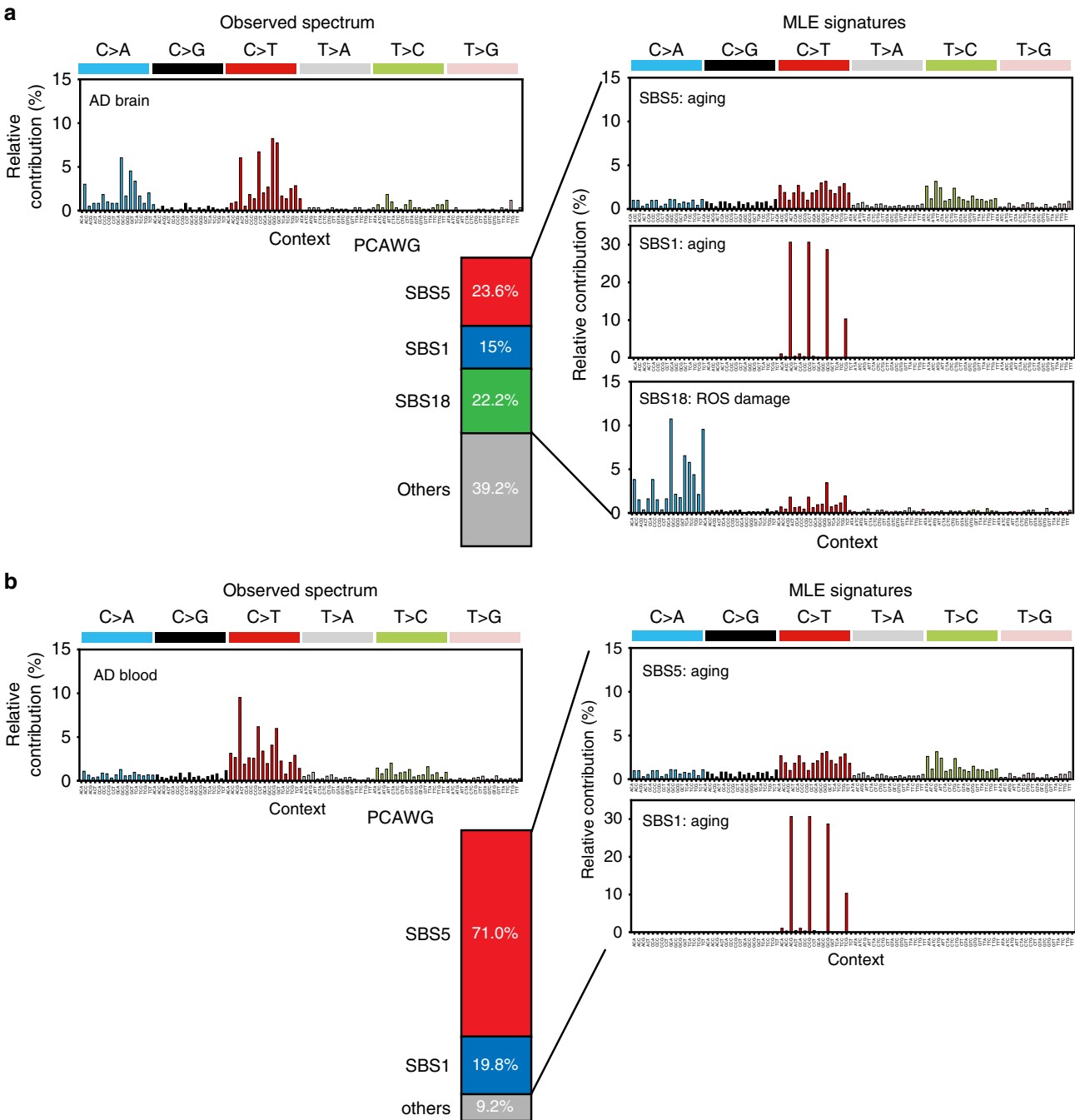

**Fig. 2** Mutation signatures of brain and blood somatic mutations in Alzheimer's disease (AD). The best decomposed mutation signature models from multiple likelihood estimation were derived for each tissue along with actual distribution of 96 possible mutation types. **a** Single base substitution (SBS) signatures 5, 1, and 18 and others account for 23.6%, 15%, 22.2%, and 39.2% of brain somatic mutations in AD, respectively. **b** SBS signatures 5 and 1, and others account for 71.0%, 19.8%, and 9.2% of blood somatic mutations in AD, respectively

we performed LCM followed by targeted amplicon sequencing on AT8-positive (a marker for the hyperphosphorylated tau) neurons in the entorhinal cortex, compared to AT8-negative neurons. We found that the VAF of the mutation, which was 1.8% in the bulk HIF tissue, was significantly enriched by 4.9-fold (VAF = 8.75%) in AT8-positive neurons ($n = 56$), whereas the mutation was not detected in AT8-negative neurons ($n = 52$) in the entorhinal cortex (Supplementary Fig. 6, Supplementary Data 10, and Methods). This result suggested that entorhinal cortical neurons with the pathogenic somatic mutation in *PIN1* are likely to be the site of origin of tau pathology.

The *PIN1* p.T152M mutation is located within the catalytic domain, and any disruption in this region may attenuate or abolish PIN1 enzymatic activity, resulting in a decrease or loss in isomerase function and subsequent disruption de-phosphorylation of tau by protein phosphatase 2 A (PP2A)[41]. To test the functional impact of the mutation, we cloned and expressed N- or C-terminal 3× FLAG-tagged wild-type and mutant human *PIN1* gene in Neuro-2a cell line and compared its messenger RNA (mRNA) and protein expression levels. No statistically significant differences were observed in relative mRNA levels of wild-type and mutant human *PIN1* (Supplementary Figs. 8a, b). Surprisingly, exogenous

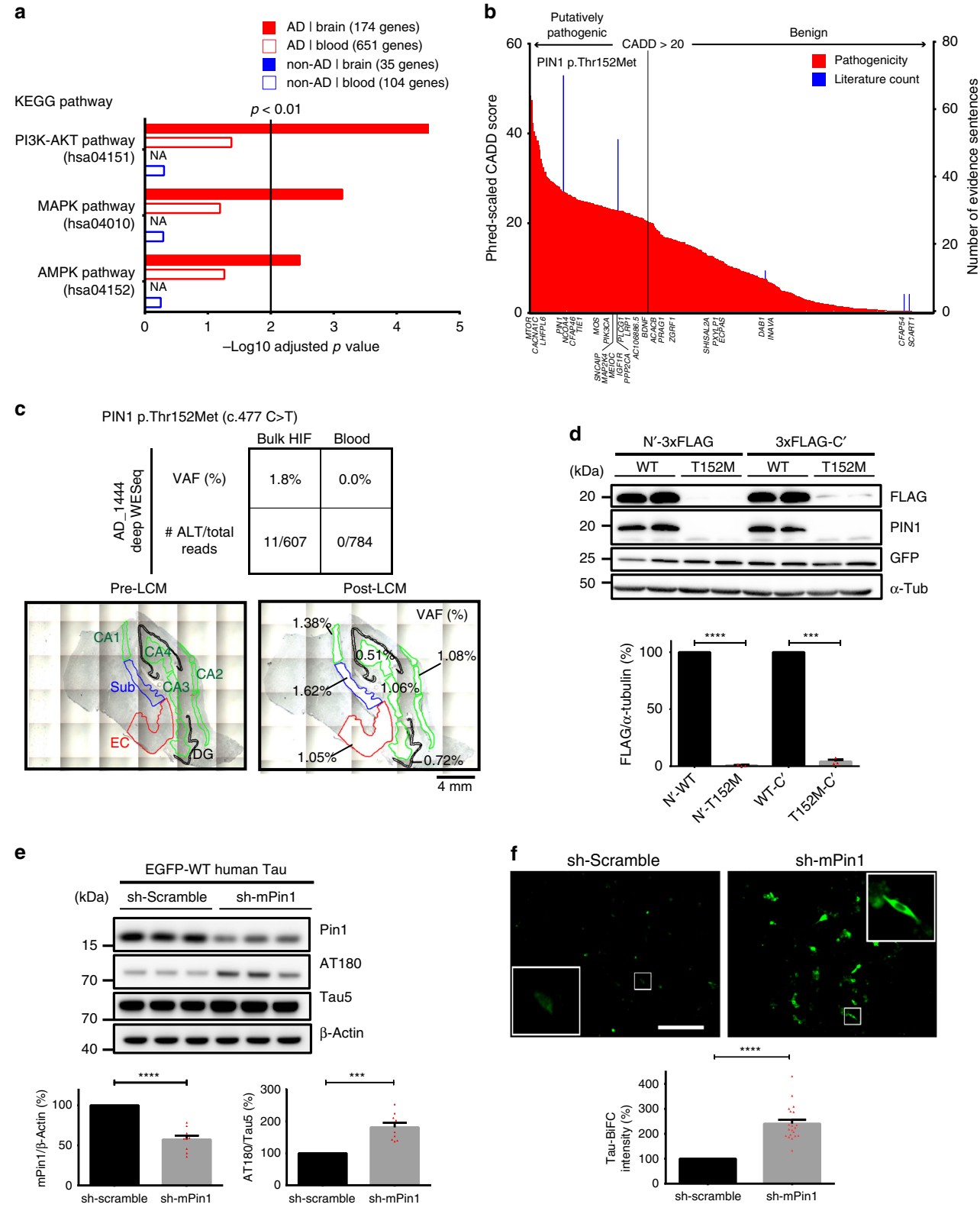

expression of the mutant PIN1 protein was almost completely lost, regardless of the location of 3× FLAG tag (Fig. 3d). Consistent with this result, protein structure-based stability prediction also indicated that p.T152M is a destabilizing point mutation (Supplementary Fig. 7).

The loss of Pin1 expression as a result of such a low-level somatic mutation in the brain could very well result in an alteration of gene dosage (i.e., expressing one half of the normal functioning protein from one copy of a wild-type allele) of *PIN1* in about 1.02–3.24% of cells in the HIF. As we suspected that disruption in *PIN1* expression could elicit dysregulation of tau phosphorylation[40,42], we further performed knockdown experiments with the human tau expressing HT22 cell line to document differences in phospho-Tau and neurofibrillary tangles level. We

**Fig. 3** Putatively pathogenic brain somatic mutations associated with tau pathology in Alzheimer's disease (AD). **a** Gene-list enrichment test of putatively pathogenic somatic mutations using the KEGG (Kyoto Encyclopedia of Genes and Genomes) pathway database. Genes with putatively pathogenic brain somatic mutations are significantly overrepresented in PI3K-AKT, mitogen-activated protein kinase (MAPK), and AMP-activated protein kinase (AMPK) pathways related to tau hyperphosphorylation. Vertical bar represents threshold for Benjamini–Hochberg adjusted P value. **b** Pathogenic scoring and text-mining results of brain somatic mutations found in AD. *PIN1* was most frequently mentioned in relevant biomedical literature and ranked 52th according to Phred-scaled CADD (combined annotation-dependent depletion) score. **c** Variant allelic frequency (VAFs) of *PIN1* c.477C>T (p.Thr152Met) in blood, bulk hippocampal formation (HIF), and each sub-region of the HIF. *PIN1* c.477C > T (p.Thr152Met) were observed. The VAF in each sub-regions of the HIF was 0.51–1.62%. **d** Comparison of protein expression levels of wild-type and mutant PIN1. PIN1 expression was analyzed in Neuro-2a cells expressing wild-type and the mutant 3× FLAG-hPin1. $n = 3$ for each experiment. **e** Effect of Pin1 knockdown on hyperphosphorylation of tau. Expression levels of pThr231-tau were observed in scramble and mPin1-shRNA (short hairpin RNA)-transfected HT22 cells. $n = 9$ for each experiment. **f** Effect of Pin1 knockdown on the oligomerization of tau. HT22 cells were co-transfected with tau-BiFC and either scramble or mPin1-shRNA. Then, cellular responses of tau-BiFC fluorescence (green) were measured. $n = 20$ for each experiment. Scale bar, 250 μm. Each bar represents the mean ± SEM and paired $T$ test were used to determine the significance of each experiment in **d**–**f**. ***$p < 0.001$; ****$p < 0.0001$. Source data is available as a Source Data file

screened three different short hairpin RNAs (shRNAs) targeting murine Pin1 and selected one shRNA that reduced mPin1 expression by nearly 50% (Supplementary Figs. 8c, d). Interestingly, when mPin1 expression was halved by the shRNA, phosphorylation levels at the direct Pin1 binding site on tau (phospho-Thr231) increased up to 1.8-fold (Fig. 3e). To further investigate tau pathology, we tested whether such haplo-insufficient expression of Pin1 is sufficient enough to trigger oligomerization of tau through BiFC assay[43]. When HT22 cell lines were co-transfected with tau-BiFC construct and mPin1 shRNA, oligomeric Tau increased up to 2.4-fold in the cells, compared to cells treated with scramble shRNA (Fig. 3f and Supplementary Fig. 9). Taken altogether, these results suggested that putatively pathogenic brain somatic mutations in AD individuals are implicated in the hyperphosphorylation and aggregation of tau protein.

**Landscape of pathogenic germline and somatic mutations in AD.** Finally, we sought to quantify the contribution of pathogenic somatic and germline mutations to Alzheimer's disease. For germline mutations in AD risk genes, we examined base substitutions and small indels in six AD risk genes, including *APOE*, *APP*, *MAPT*, *PSEN1*, *PSEN2*, and *TREM2*, and 290 known pathogenic SNPs therein from the AlzGene mutation database[44] (Supplementary Data 11 and Methods). We found that 15.4% (8/52) and 48.1% (25/52) of AD individuals carried two copies or one copy of *APOE* ε4 alleles, respectively. We found that one patient carried an AD risk modifier in *PSEN1* (AD_1447, p. E318G), while three patients and one control exhibited AD risk modifiers in *TREM2* (AD_318, AD_191, and non-AD_203, p. R62H; AD_1451, p.R47H). Interestingly, 26.9% (14/52) of AD individuals had at least one putatively pathogenic brain somatic mutation associated with biological pathways affecting tau phosphorylation (Fig. 4a). Consistent with previous findings[45–47], individuals expressing pathogenic germline risk factors (67.33 ± 2.09, $n = 12$) were significantly younger at AD onset than non-carriers (75.3 ± 2.1, $n = 33$) (Supplementary Fig. 10c). Among 14 AD patients with putatively pathogenic somatic mutations in tau phosphorylation-modifying pathways, five also had two copies of *APOE* alleles. Overall, while only 13.5% (7/52) of AD patients could be explained by germline mutations alone, 17.3% (9/52) and 9.6% (5/52) of the AD patients in our cohort carried somatic mutations alone and both germline and somatic mutations, respectively (Fig. 4b).

**Discussion**

The molecular genetic mechanism of how tau aggregates are initiated from the entorhinal cortex and hippocampal area has remained a long-standing question. In the present study, using

comprehensive investigation of low-level somatic mutations in HIFs of AD patients, followed by experimental functional studies, we showed that brain somatic mutations accumulating with increasing age can modulate the initial appearance of tau pathology in the HIF of AD brains. Although increasing evidence has shown that low-level brain somatic mutations are crucial in various neurological disorders[48], it remains unclear whether aging-associated somatic mutations can affect molecular pathogenesis, mainly due to technical constraints and difficulties in unbiased identification of low-level pathological brain somatic mutations at a genome-wide level. For example, single-cell sequencing technology recently revealed that individual neurons from the prefrontal cortex and dentate gyrus accumulate ~23 and ~40 somatic SNVs every year[24]. Putting aside errors arising from whole-genome amplification or clonal cell expansion[20], somatic mutations unique to an individual cell are unlikely to explain the molecular genetic pathogenesis underlying a given neurological disorder. However, our study used deep whole-exome sequencing of micro-dissected HIF tissue to identify low-level brain somatic mutations contributing to the initiation of tau pathology in AD brains.

Our data suggest that 22 and 106 somatic SNVs would accumulate in HIF and blood tissues, respectively, every year. The VAF and number of somatic mutations in the blood were significantly higher than those in the brain. Mature blood and immune cells are produced by the process of hematopoiesis. As people age, clonal expansion of mutated stem cells within the blood occurs and is associated with increased numbers of somatic mutations and a greater risk of developing hematological malignancies[49]. Consistent with this, we also observed that blood cells have significantly higher numbers and distinctive patterns of subclones relative to brain cells (Supplementary Fig. 11 and Methods). Together with mutation signatures of somatic mutations in blood, this suggests that clonal hematopoiesis contributes to the increased number and VAF of somatic mutations in a patient's blood.

Previous studies have highlighted abnormal increases in the activity of MAPK[50] and decreases in the activity of AMPK[51] and PP2A[52] in post-mortem AD brains, compared to control brains of individuals not affected by AD. An imbalance between such tau kinases and phosphatase activity causes hyperphosphorylation and aggregation of tau, which subsequently affects synaptic plasticity and memory impairment in AD[53,54]. According to the structure-based protein stability prediction, we were able to identify putatively pathogenic brain somatic mutations that could be expected to be destabilizing mutations (Supplementary Data 12). For instance, unstable expression of subunits (*PPP2CA* and *PPP2R1A*) of the primary tau phosphatase PP2A[52,55] could modulate the function of tau phosphatase and subsequently induce hyperphosphorlyation of Tau[56]. Changes in

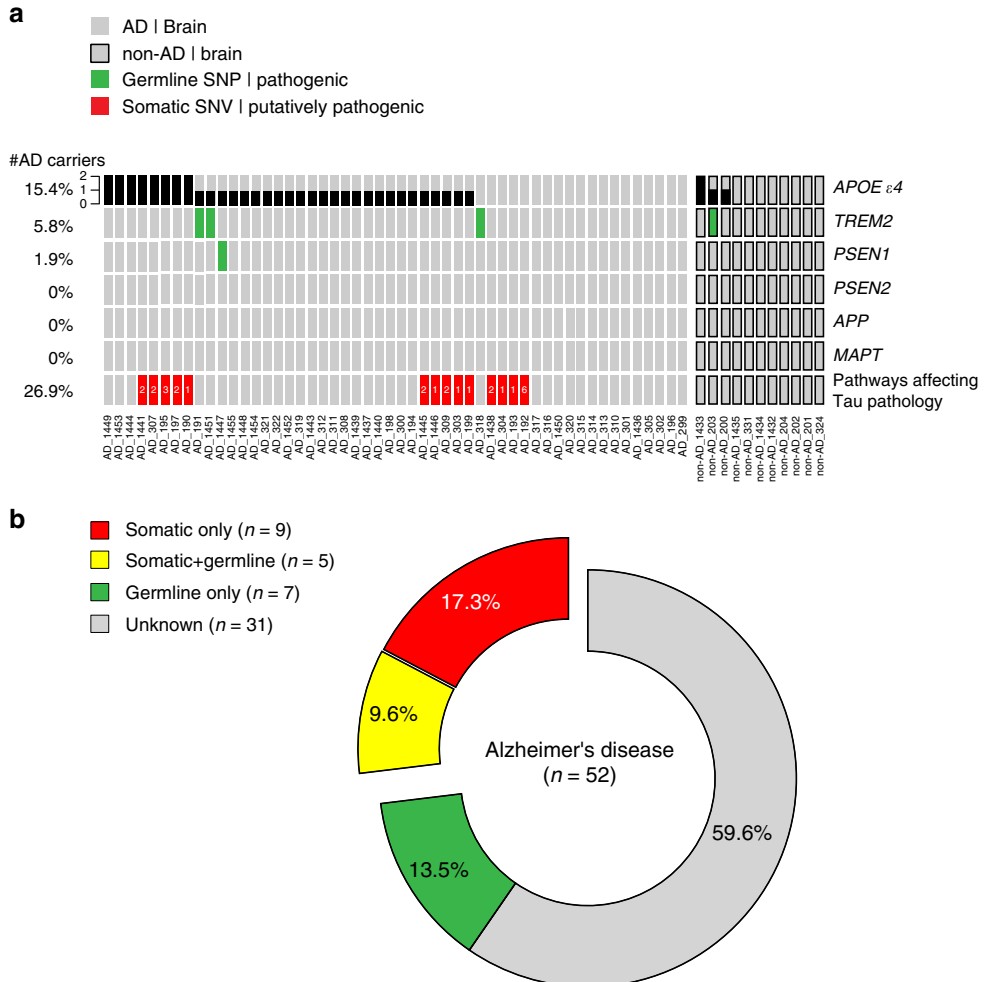

**Fig. 4** Landscape of pathogenic germline and somatic mutations contributing to Alzheimer's disease (AD). **a** The presence of pathogenic germline risk factors and putatively pathogenic somatic mutations across all AD and non-AD individuals. The copy number of *APOE* ε4 alleles are annotated in black. Known risk modifiers of AD in *APP*, *PSEN1/2*, *TREM2*, and *MAPT* are marked in green. Putatively pathogenic brain somatic mutations associated with tau hyperphosphorylation-related pathways are marked in red, and the number of affected genes was annotated together. **b** Categorization of pathogenic germline and somatic mutation carriers among AD patients. 13.5% (7/52) of AD patients in our cohort could be explained by germline mutations alone; 26.9% (14/52) of AD patients harbored putatively pathogenic somatic mutation affecting the phosphorylation of tau

the protein stability of the MAPK-specific inhibitor *PTPN5*, which is predominantly expressed in neurons, would favor an increase in a tau kinase MAPK activity[57,58]. Meanwhile, a destabilizing mutation in PI3K activators (*HGF*, *ITGB4*)[59,60] or the catalytic domain of PI3K (*PIK3CA*)[61] would also likely increase tau phosphorylation through deactivating AKT, a negative regulator of the primary tau kinase glycogen synthase 3β (GSK-3β)[62,63].

Meanwhile, there have been recent studies of brain somatic mosaicism in sporadic Alzheimer's disease[64,65]. Although these studies used targeted amplicon sequencing or low-depth exome sequencing, we could not detect any of the AD-associated somatic mutations reported in therein in our cohort (Supplementary Data 13). However, we were able to observe 1–17 intra-exonic junctions in the *APP* gene from four different HIFs of AD patients, which were reported by the recent study of *APP* genomic complementary DNAs in AD[66] (Supplementary Fig. 12).

According to our mutation signature analyses, 22.2% of brain somatic mutations in AD are associated with ROS-mediated changes in DNA. Consistent with these findings, previous studies have reported that excessive generation of ROS by mitochondrial dysfunction and ROS-mediated oxidative DNA damage are

central features of neurodegenerative diseases, including AD and Parkinson's disease (PD)[3,67]. Higher metabolic demands in neuronal cells make the brain more susceptible to oxidative stresses[68]. Indeed, research has shown that neurons in the entorhinal cortex[69] and CA1 region[70] are more sensitive to oxidative stress, compared to other brain regions. Increased protein oxidation has also been detected in the hippocampus of AD patients and the substantia nigra of PD patients[71,72]. As mitochondrial damage could lead to an excessive generation of ROS and insufficient ATP production, the accumulation of somatic mutations in mitochondrial DNA has been thought to be the key driver for age-related neurodegeneration[73]. Several studies have observed a significant increase of mitochondrial mutation burden in hippocampal neurons in early-stage AD and in substantia nigra neurons in early-stage PD patients[74,75]. Interestingly, 55.6% (15/27) of the putatively pathogenic somatic mutations that we identified in tau phosphorylation pathways accumulated as G:C → T:A transversion mutations, a hallmark of oxidative DNA damage[76,77] (Supplementary Data 12).

Although follow-up studies will be necessary to document recurrence of the brain somatic mutations in larger cohorts and to causally link them to AD pathogenesis, our study provides new

insights into the molecular genetic etiology of AD and other sporadic neurodegenerative disorders potentially linked to somatic mutations in the brain.

## Methods

**Tissue collection.** Fresh frozen human hippocampus and matched whole blood samples were generously provided by two brain banks (Netherlands Brain Bank [NBB] and Human Brain and Spinal Fluid Resource Center [HBSFRC]). Samples were obtained from clinically and neuropathologically classified AD-affected individuals, as well as age-matched unaffected controls. NBB uses NINCDS-ADRDA criteria and HBSFRC uses CERAD criteria for neuropathological diagnosis. Control cases were defined as those in which a Braak neurofibrillary tangle staged 1 or less, and there were no other features suggestive of neurodegenerative disease in either ante mortem inspection or post mortem assessment. Genomic DNA (gDNA) of brain and peripheral tissues (liver or spleen) from 21 healthy controls and 11 schizophrenia cases were provided from the Stanley Medical Research Institute. The NBB, HBSFRC, and SMRI obtained permission from the donors for brain autopsy and use of tissue, blood, and clinical information for research purposes. The study was performed with informed consent according to protocols approved by Institutional Review Boards of KAIST, as well as the Committee on Human research (IRB #: KH2014-36).

**gDNA extraction.** gDNA was isolated from 6 to 10 Nissl-stained tissue slides by LCM. Briefly, a frozen hippocampal tissue block was cryosectioned at 20 μm by cryostat (Leica, CM1850) and attached to ultraviolet (UV)-treated 1.0 mm PEN-membrane slides (Zeiss, 415190-9041-000). Then, each slide was stained with 1% Cresyl violet-75% EtOH solution right before LCM. After confirming that subregions of the HIF were properly stained, slides were mounted on the stage of an LCM machine (Zeiss, PALM MicroBeam). HIFs were captured from each slide and kept in lysis buffer (Qiagen, 56304) throughout the acquisition. Acquired tissue was mechanically crushed by bead-bitting homogenizer (MP Biomedicals, FastPrep-24). Then, the homogenized tissue was further lysed in 56 °C for 12 h. Digested samples were purified using a column-based QIAamp DNA Micro Kit (Qiagen, 56304). Whole blood cells were processed using QIAamp DNA Blood Midi Kit (Qiagen, 51183) following the manufacturer's instructions. Extracted gDNA was quantified using Bioanalyzer 2100 (Agilent, USA), and integrity was checked by running on 1% agarose gel. The average yield of gDNA from HIF was calculated by measuring the volume of LCM-captured region from the PALM Robo software and the concentration of gDNA from Bioanalyzer. For the estimation of the number of neurons from LCM-captured region, we firstly counted the average number of observed neurons in each sub-region of HIF by finding number of maximum signals from 8-bit converted Nissl-stained images with the Image J software. Then, we multiplied the area of LCM-captured region and the aforementioned average neuron counts.

**Deep whole-exome sequencing.** Each exome library was prepared by following the manufacturer's instructions (Agilent, Human All Exon V4/V5 + UTR 50 Mb Kit) using ~1 μg of gDNA as input. Then, we performed paired-end sequencing on an Illumina HiSeq 2000/2500 instrument (average depth 584×) according to the manufacturer's protocol using QC-passed exome libraries. We followed the GATK Best Practices (v3.5) workflow to generate analysis-ready bam files from QC-passed Fastq files. Briefly, Fastq files were aligned to reference genome (GRCh38) using BWA to generate bam files, and PCR duplicates were marked by Picard. Next, reads nearby indels in these bam files were realigned using RealignerTargetCreator and IndelRealigner from GATK analysis tools. Finally, we performed recalibration of base quality score with BaseRecalibrator from GATK analysis tools for subsequent accurate variant calling.

**Somatic SNVs calling.** Somatic SNVs were detected with MuTect[19] (v1.1.7) from the 32 matched brain–liver samples (e.g., Panel of Normals and Test dataset), 48 matched brain–blood samples, and 15 brain-only samples. We used default options, except releasing contamination fraction (fraction_contamination) to 0.0 (default = 0.02) to secure low-level somatic mutations (VAF <5%). We double checked possible foreign DNA and sample cross-contamination by running Veccum and ContEst, respectively. From the MuTect output from matched samples, we filtered out possible unreliable calls by applying the following criteria: (i) excluding those with <35 total read depths; (ii) excluding those with VAF ≥40% as suspected germline mutations; (iii) excluding those with an EBscore ≤2.396; (iv) excluding variants with all supporting reads located at either end of reads; (v) manual inspection with IGViewer (v2.3.94). For the manual inspection, we checked the following: (a) supporting reads with altered alleles had no other base changes, unless they were heterozygotic/homozygotic SNPs; (b) the average of second highest BLAT score of supporting reads was <900; (c) more than 50% (at least three reads) of supporting reads were secured. For brain-only samples, a more strict depth (<100 are excluded) and VAF (≥20% are excluded) were applied. All putative SNVs were annotated with VEP for characterization of mutation subtypes using the following six simplified categories: (a) Non-synonymous—missense variant, stop gained, start lost, stop lost, splice donor, or splice acceptor; (b) Synonymous—synonymous variant; (c) UTR—5′ prime UTR variant or 3′ UTR variant; (d) intronic—intron variant, mature miRNA variant, non-coding transcript exon variant, or stop retained variant; (e) splicing—splicing region variant; and (**f**) intergenic—intergenic variant, upstream variant, or downstream variant. Phred-scaled CADDscore (v1.4, GRCh38 model) was used for pathogenicity scoring of variants and a cut-off of >20 was applied for demarcating putatively pathogenic variants. For discerning rare variants, minor allele frequency (all exomes) of each variant was collected from gnomAD[31], and a cut-off of <0.01% was applied.

**Empirical Bayesian score.** To apply EBscore[22] on whole call-set, we initially checked the performance of the algorithm with a test dataset. We sequenced and processed matched brain and peripheral samples from 21 healthy controls and 11 schizophrenia cases identical to AD and non-AD samples. Then, by randomly choosing 54 somatic SNVs, we performed targeted amplicon sequencing to distinguish true calls from false calls. On each variant position, EBscore was applied using 21 healthy controls as a panel of normal samples. A receiver operating characteristic (ROC) curve was made using the pROC R package, and confirmed that the area under curve was high enough (>0.9). An optimal cutoff point was determined when the sum of sensitivity and specificity was maximized.

**Targeted amplicon sequencing.** Primers were designed with the Primer3 algorithm and were synthesized from Macrogen (Seoul, Korea). Amplicons were prepared by two-step PCR using Illumina TruSeq adapters. First, PCR reactions were carried out using 10 ng of initial template gDNA. Then, 1st-amplicons were analyzed on 2% agarose gels and the bands with expected size were isolated and purified using Mega Quick-Spin Kit (iNtRON, Korea). Next, 100 ng of purified 1st-amplicons were used as templates for second PCR, and the products were subsequently purified with the aforementioned purification kit. 2nd-Amplicons were quantified using Bioanalyzer 2100 (Agilent, USA). The QC-passed amplicons were sequenced on a Hiseq 2500 (Illumina, USA) sequencer. Generated Fastq files were aligned to GRCh38 reference genome by BWA-MEM, and reads on target sites were filtered for MQ20 and BQ30 with the bam-readcount R package. We double checked the number and quality of altered alleles by visualization with IGViewer (v2.3.94). For estimation of the background error rate, we previously constructed spike-in samples and performed replicate sequencing. If such a mutation was present as statistically reliable compared to previously estimated background errors, we considered using them as true calls. The background error rates of the PCR-based amplicon-based platforms were as follows: T > A (VAF=0.00312107), T > C (VAF=0.007970457), T > G (VAF=7.58E − 04), C > T (VAF=0.004071926), C > G (VAF=7.65E-04), and C > A (VAF=0.001847634)[78].

**Targeted amplicon sequencing on AT8-positive neurons.** A freshly frozen hippocampal tissue block was cryosectioned at 20 μm by cryostat (Leica, CM1850) and attached to UV-treated 1.0 mm PEN-membrane slides (Zeiss, 415190-9041-000). Then, each slide was fixed in phosphate-buffered 4% paraformaldehyde for 15 min at room temperature and subsequently washed in phosphate-buffered saline (PBS). The tissue slides were blocked with PB-GT (0.2% gelatin, 0.2% Triton X-100 in PBS) for 1 h at room temperature. Then, the slides were treated with 1:500 anti-PHF-tau [AT8] (Thermo, MN1020) and 1:500 NeuN (Abcam, ab104225) and incubated overnight at 4 °C. On the next day, the slides were washed two times with PBS for 10 min each. After washing, the slides were further treated with Alexa Fluor 488-conjugated goat antibody to mouse (Thermo, A11001) and Alexa Fluor 594-conjugated goat antibody to rabbit (Thermo, A11012) for 1 h at room temperature. After incubation, the slides were washed two times with PBS for 10 min each. From the freshly stained slides, AT8-positive neurons (n = ~50) were microdissected with the PALM laser capture system and collected in an adhesive cap (Zeiss, 415190-9201-000). gDNA was extracted from the collected neurons using QIAamp DNA Micro Kit (Qiagen, 56304) according to the manufacturer's protocol. Then, gDNA was used for validating PIN1 c.477C>T mutation. Primers for the target site were designed with the Primer3 algorithm and were synthesized from Macrogen (Seoul, Korea). The detailed sequence of the primer set was as follows: forward; 5′-AGACGCCTCGTTTGCGCTGC-3′ and reverse; 5′-GGGG TTCGGCCACTGGCTGG-3′. Amplicons were prepared as mentioned in the previous section and then QC-passed amplicons were sequenced on an Illumina Miseq sequencer (SoVarGen, Korea).

**Mutation signature analysis.** To determine the contribution of mutation signatures, we pooled somatic SNVs from all AD subjects and divided into two groups, brain and blood. Then, we formatted each pooled SNVs in VCF files and used them as input files for running Mutalisk[28]. The following options were used (MLE method: linear regression). The input files were compared with exome-adjusted PCAWG reference signatures based on multiple likelihood estimation method followed by constraining linear function. The best model of signature combinations for each group was suggested from the tool by considering both Cosine similarity and Bayesian information criterion.

**Text-mining biomedical literatures using DigSee.** To identify candidate genes associated with both Alzheimer's disease and dysregulation of tau phosphorylation, we used the all-disease version of DigSee[38], a text-mining based search engine for

disease–gene relationships. Gene symbols were pooled from AD brain somatic mutations and the list of the genes was used as input file for running DigSee. The number of evidence sentences having both "Alzheimer's disease" and "Phosphorylation" for each gene was counted.

**Protein stability prediction**. To predict the impact of putatively pathogenic somatic mutations on protein stability, we used the knowledge-based protein stability estimation tool, SDM2. Given a PDB ID and list of point mutations, the SDM2 server calculates the difference in stability scores ($\Delta\Delta G$) between wild-type and mutant protein.

**Gene-set enrichment analysis with Enrichr and DNENRICH**. We separately collected all genes with putatively pathogenic somatic mutations in brain and blood tissues from AD patients and controls unaffected by AD (AD_Brain, AD_Blood, non-AD_Brain, non-AD_Blood). Then, we performed gene-set enrichment analysis by Enrichr[33] to identify critical biological processes in which putatively pathogenic variants were overrepresented. The KEGG database (v.2016) was used as a reference database to find AD-relevant pathways. To exclude gene length bias in enrichment test, we independently performed gene length adjusted gene-set enrichment test with the DNENRICH algorithm. Briefly, we separately collected all genes with putatively pathogenic somatic mutations in brain and blood tissues from Alzheimer's disease patients and controls unaffected by AD (AD_Brain, AD_Blood, non-AD_Brain, non-AD_Blood). Then, we performed gene-set enrichment analysis by DNENRICH with 100,000 permutations to identify critical biological processes in which pathogenic variants were overrepresented.

**Random permutation test**. We collected all genes with putatively pathogenic somatic mutations in brain and blood tissues of AD patients and controls unaffected by AD (AD_Brain, AD_Blood, non-AD_Brain, non-AD_Blood), which totaled 930 unique gene pools in our cohort. Then, we utilized the reference gene list of the aforementioned Tau pathology associated pathways (PI3K-AKT, MAPK, and AMPK pathways) from the KEGG database (v. 2016). To simulate the AD_Brain gene set overlapping each of the reference gene sets, we randomly selected 174 genes (the same number of genes from AD_Brain) from our cohort gene pools with 10,000 permutations (random re-sampling). From the permutation distribution of overlapping genes in the reference gene sets, we estimated significance by comparing enrichment test overlap counts (PI3K-AKT: 15; MAPK: 11; AMPK: 7) for AD_Brain gene sets with a distribution cutoff of 5%.

**Cloning, mutant construction, and expression of Pin1**. pCIG-human PIN1 was generated by inserting human PIN1 complementary DNA (cDNA) from GST-PIN1 (Addgene, 19027) into Xho1/EcoR1 sites of pCIG2-C1 (modified from pCIG2). Met mutant of hPIN1 at Thr-152 (T152M) was generated using hPIN1 as a template with a QuickChange XL Site-Directed Mutagenesis Kit (Agilent, 200516). The following primers were used for the mutagenesis: T152M sense, 5′-GCCGGAATCCATGAACACGGGCCCG-3′ and T152M anti-sense, 5′-CGGG CCCGTGTTCATGGATTCCGGC-3′. To append 3×FLAG tag on either the N,C-terminals of hPIN1 and T152M mutant, we synthesized 3×FLAG DNA fragments by annealing the following oligos: (i) N′-3×FLAG sense, 5′-ATGGACTAC AAAGACCATGACGGTGATTATAAAGATCATGACATCGACTACAAGGATG ACGATGACAAG-3′ and N′-3×FLAG anti-sense, 5′-CTTGTCATCGTCATCCT TGTAGTCGATGTCATGATCTTTATAATCACCGTCATGGTCTTTGTAGTCC AT-3′ (ii) 3×FLAG-C′ sense, 5′-GACTACAAAGACCATGACGGTGATTATAA AGATCATGACATCGACTACAAGGATGACGATGACAAGTAG-3′ and 3×FLAG-C′ anti-sense, 5′-CTACTTGTCATCGTCATCCTTGTAGTCGATGTCA TGATCTTTATAATCACCGTCATGGTCTTTGTAGTC-3′. Then, we performed overlap PCR with 3×FLAG and hPIN1 cDNA as templates and assembled them with another linear DNA fragment prepared from Xho1/EcoR1-digested pCIG2-C1 plasmid using an EZ-Fusion Cloning Kit (Enzynomics, EZ016S) to obtain pCIG-3×FLAG-hPIN1. For quantitative reverse transcription PCR (RT-qPCR), 1 μg of 3×FLAG-hPIN1 WT and T152M mutant plasmid DNA were transiently transfected in Neuro-2a cell line (ATCC, CCL-131) and cells were harvested after 24 h of transfection. Meanwhile, for protein expression test, 3 μg of 3×FLAG-hPIN1 WT, and T152M mutant plasmid DNA were transiently transfected in Neuro-2a cell line and cells were harvested after 72 h of transfection.

**Quantitative reverse transcription PCR**. Transfected cells were collected by scrapping with cold 1× PBS and total RNA was extracted using RNA-spin Total RNA Extraction Kit (iNtRON, Korea) according to the manufacturer's instructions. The quantity and quality of RNA were measured with a NanoDrop Lite (Thermo, USA) and subsequently confirmed that total amounts of RNA were not significantly different between the samples. For cDNA synthesis, 500 ng of total RNA was reverse transcribed into cDNA using the ReverTra Ace qPCR RT Master Mix with gDNA remover (TOYOBO, Japan) according to the manufacturer's instructions. The real-time monitoring of PCR amplification reaction was performed on a CFX-96 Touch Real-Time PCR Detection System (Bio-Rad, USA) using SYBR Green Real-Time PCR Master Mix (TOYOBO, Japan) according to the manufacturer's instructions. Thermal cycling of a three-step real-time PCR protocol with a melting-curve analysis step was conducted as follows: 95 °C for 1 min, followed

by 39 cycles of 95 °C for 15 s, 58 °C for 20 s, and 72 °C for 30 s in sequence. Each sample was run in triplicate on separate wells to allow for 3×FLAG and human PIN1 quantification relative to Gapdh. The primers used for the RT-qPCR are listed in Supplementary Fig. 8b. The relative fold changes of the mutant constructs, compared to wild-type PIN1, were calculated using the ΔΔCt method.

**Western blot**. Transfected cells were collected by scrapping with cold 1× PBS and subsequent protein lysates from cells were prepared with 1% Triton X-100 in PBS with EDTA and Halt protease and phosphatase inhibitor cocktail (Thermo, 78440). After lysis for 30 min in 4 °C, the lysates were centrifuged at $16,200 \times g$ for 30 min and supernatants were used to measure protein concentration with BCA assay Kit (Thermo, 23225). Harvested proteins were resolved by sodium dodecyl sulfate-polyacrylamide gel electrophoresis and transferred to PVDF membranes (Millipore, IPVH00010). Transferred membranes were blocked with 5% bovine serum albumin (BSA) (Bovogten, BSAS 0.1) in TBS, which contains 0.1% Tween-20 (TBST) for 1 h at room temperature. After the blocking, the membranes were incubated with primary antibodies, including 1:1000 anti-α- tubulin (Cell Signaling, 3873), 1:1000 anti-FLAG M2 (Cell Signaling, 8146), 1:500 anti-Pin1 (Santa Cruz, 46660), 1:2500 anti-GFP (Abcam, ab290), 1:1000 tau5 for total tau (Thermo, AHB0042), 1:1000 AT180 for phosphor-tau (Thr231) (Thermo, MN1040), and 1:2000 anti-β-actin (Cell Signaling, 3700), in TBST for overnight in 4 °C. Then, the next day, the membranes were serially washed with TBST for four times. After washing, the membranes were then further incubated with either 1:10,000 horse-radish peroxidase linked anti-rabbit or mouse secondary antibodies (Cell Signaling, 7074/7076) for 2 h at room temperature. After the incubation followed by another four times of washing with TBST, immune detection was performed using ECL substrates (Bio-Rad, 1705060) on ChemiDoc XRS + imaging system (Bio-Rad, USA).

**Knockdown of murine Pin1**. Three different pLKO.1-murine Pin1 shRNAs were designed from suggested target sites by GPP Web portal. Target-specific scramble shRNA was designed by random shuffling one of the target sequence of the murine Pin1 targets. HT22 cells (gifted from Dr. David Schubert, Salk Institute) were co-transfected with 2 μg of murine Pin1 targeting shRNA (or scramble) and 1 μg of EGFP-WT human tau using Lipofectamine LTX with Plus reagent (Thermo, 15338100) according to the manufacturer's instructions. After 48-h transfection, protein levels were measured by Western blot.

**BiFC assay for tau aggregation**. HT22 cells were co-transfected with mPin1 shRNA (or scramble) and tau-Venus BiFC construct using Lipofectamine LTX and Plus reagent (Thermo, 15338100). tau-Venus BiFC construct was constructed from a mammalian bidirectional expression vector (pBI-CMV1). Briefly, hTau-WT-4R2N-VC155 from the pCMV6-AC-hTau-WT-4R2N-VC155 (i.e., pCMV6-Tau40-VC155) plasmid was cloned at the MCS2 EcoRI site in pBI-CMV1 vector. Then, hTau 4R0N isoform was generated using domain deletion PCR. Next, hTau-WT-4R0N-VN173 from the pCMV6-AC-hTau-WT-4R0N-VN173 plasmid was inserted into MCS1 using BamHI and SalI sites. After transfection, puromycin selection was followed every 24 h for 2 days. tau-BiFC signals were captured using an EVOS™ FL Auto 2 Imaging System (Thermo, USA), and the fluorescence intensities were analyzed by Image J software.

**Pathogenic germline mutations in AD-risk genes**. In total, 290 known AD pathogenic mutation sites (e.g., pathogenic, risk modifier, possible risk modifier) in three autosomal dominant (APP: 28 sites, PSEN1: 239 sites, PSEN2: 16 sites) and two other AD-associated genes (TREM2: 6 sites and MAPT: 1 site) were curated from the AlzGene mutation database (last updated on 27 April 2018)[44]. In addition, individuals carrying two copies of APOE ε4 (rs429358 [C], rs7412 [C]) were also considered as having pathogenic germline mutations. We used the bam-readcount R package to quantify the number of reference and alternative alleles in brain samples at each germline SNP site. To ensure germline SNPs, we used filtered reads (MQ20, BQ30) and only considered germline SNPs as those with a VAF ≥40%. For those suspected heterozygous germline SNPs observed from the bam-readcount result were further verified with GATK HaplotypeCaller (v3.5).

**Clonal reconstruction with LICHeE**. We used the somatic SNV-only mode to perform VAF-based clustering and subsequent clonal lineage reconstruction according to the author's guideline with modification of the following parameters to account for delineating clusters with low-level somatic mutations: -maxVA-FAbsent 0.005, -minVAFPresent 0.005, -minClusterSize 1, -maxClusterDist 0.005. The best-scored linage tree in each sample was exported as DOT format for Graphviz visualization.

**Visualization of genomic complementary DNAs in APP gene**. Raw fastq files were aligned to the human reference genome (GRCh38) using STAR aligner with following settings: --outSAMattributes All, --outFilterScoreMinOverLread 0.8, --outSJfilterCountTotalMin 1 1 1 1. Duplicate reads were marked and removed by Picard and Bam files were then filtered to only include lines without "jI:B:i,-1" tag,

which indicates no junction is detected. Resulting Bam files were then converted to bed12 format to be visualized in exonjunction R package.

**URLs**. Variant Effect Predictor (VEP), https://asia.ensembl.org/Tools/VEP; Mutation Signature Analyses (Mutalisk), http://mutalisk.org/analyze.php; Disease Gene Search Engine (DigSee), http://gcancer.org/digsee; Site Directed Mutator (SDM2), http://marid.bioc.cam.ac.uk/sdm2; Gene-set enrichment analysis (EnrichR), http://amp.pharm.mssm.edu/Enrichr; AlzGene mutation database, https://www.alzforum.org/mutations; GPP Web portal for shRNA targets, https://portals.broadinstitute.org/gpp/public/gene/search;

Designing scramble shRNA, https://www.genscript.com/tools/create-scrambled-sequence;

Combinatorial method for reconstructing cell lineage trees and inferring sub-clonal composition (LICHeE), https://github.com/viq854/lichee;

A framework for calculating recurrence and gene-set enrichment with gene length adjustment (DNENRICH), https://psychgen.u.hpc.mssm.edu/dnenrich/;

Visualizing gencDNAs in APP gene from genomic reads, https://github.com/christine-liu/exonjunction.

**Reporting summary**. Further information on research design is available in the Nature Research Reporting Summary linked to this article.

## Data availability

All datasets generated and/or analyzed during the current study are presented in this article or the accompanying Source Data or Supplementary Information files will be available from the corresponding authors upon request. 111 Deep whole-exome sequencing data produced in the current study have been deposited in the NCBI Sequence Read Archive (SRA) with accession number PRJNA532465. Targeted amplicon sequencing data for randomly chosen validation process and confirmation of putative pathogenic variant of PIN1 (c.477C>T) are deposited in the SRA with accession numbers PRJNA532989 and PRJNA532992, respectively. The source data underlying Figs. 1c–g, 3a, 3d–f and Supplementary Figs. 1c, 4b–d, 7a, 8a–c, 10a–c, 11a and 12a–b are provided as a Source Data file.

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

## Acknowledgements

We thank H. Heinsen and Y.M.P. for guiding delineation of the sub-regions of the HIF. We appreciate Dr. L. Alexandrov (UCSD) for discussing and sharing reference datasets for mutation signature analyses. We also appreciate the IBS Center for Cognition and Sociality for permission to use their PALM MicroBeam LCM system. This work was supported by grants from the Suh Kyungbae Foundation (G01170502) awarded to J.H.L., the Korean Health Technology R&D Project, Ministry of Health & Welfare, Republic of Korea (H15C3143) awarded to J.H.L., and the Korea Institute of Science and Technology Information (K-19-L02-C07-S01) awarded to S.J.Y. We thank the Netherlands Brain Bank (Lee-835) for providing 96 brain and matched blood samples for Alzheimer and unaffected control cases, which were supplied to J.H.L. Fifteen brain samples (supplied to J.H.L.) of Alzheimer and unaffected control cases were obtained from the Human Brain and Spinal Fluid Resource Center, VA West Los Angeles Healthcare Center, 11301 Wilshire Blvd, Los Angeles, CA 90073, which is sponsored by NINDS/NIMH, the National Multiple Sclerosis Society, and the US Department of Veterans Affairs. We would also like to thank the Stanley Medical Research Institute for supplying gDNA and whole-exome sequencing data for 32 non-schizophrenia and schizophrenia cases, which were given to J.H.L.

## Author contributions

J.S.P. and J.H.L. conceived of the idea and organized the study. J.S.P. and J.L. performed genetic analysis. E.S.J. and I.M-J. performed in vitro assays of tau protein. M-H.K. and S.H.K. helped analyze sequencing data from non-schizophrenia and schizophrenia brain and peripheral tissues. I.B.K. performed random permutation tests. H.J.S. and S.W.K. measured the statistical significance of targeted sequencing data. Y.M.P. helped to acquire brain and peripheral tissues from brain banks. J.S.P. and J.H.L. wrote the manuscript. S.J.Y. and J.H.L. led the project.

## Additional information

**Competing interests:** J.H.L. is co-founder of SoVarGen Inc., which seeks to develop new diagnostics and therapeutics for brain disorders. The other authors declare no competing interests.

