## [Peer Review File · Nature Communications]

Editorial Note: Tables R1 and R2 accompanying the legends on p18 are attached to this file.

Reviewers' comments:

Reviewer #1 (Remarks to the Author):

NCOMMS-18-36041

General comments:

In this manuscript, Park and colleagues tests the hypothesis that brain somatic mutations arising in the hippocampus contributes to Alzheimer's disease (AD). They applied ~600X coverage whole-exome sequencing (WES) on 111 postmortem hippocampi (HIF) and matched blood samples from 52 AD and 11 control cases. They observed age-related accumulation of variants in laser-capture enriched HIF cells, concluding that an average of 0.53 de novo SNVs/exome occur per year, and 4.8-fold more occur in blood, and extrapolate across the whole genome to conclude that 21.32 de novo SNVs/genome occur per year. Amplicon sequencing showed validation in 80.3% of variants tested, suggesting that false-discovery was minimal. Previous work (PMID 29217584) identified ~40 de novo SNVs/year in the hippocampal dentate gyrus, which is remarkably similar to Park's results, given that different methods were used to arrive at these conclusions. Mutational signatures were also similar to the previous work, pointing to aging, DNA damage, and reactive oxygen species. There were no differences in mutation counts nor genomic location of the somatic mutations in AD vs. non-AD tissues, but AD brains showed enrichment for mutations in signaling pathways, occurring in 26.9% of AD individuals. This was one of the most exciting findings of the paper but the authors do not probe it further. The authors found a mutation in the proline isomerase PIN1, in patient AD-1444, demonstrate that it is a destabilizing point mutation, and functionally demonstrate that shRNA knockdown of Pin1 leads to increased Tau phosphorylation and oligomerization.

The methodology for identifying and validation somatic variants are reliable, and statistical analysis and benchmark are well performed. The manuscript is well-written and easy to understand. Compared with existing targeted sequencing based results, the manuscript provides a comprehensive assessment of somatic mutations and confirmed previous predictions. The manuscript sheds light on the potential for somatic mutations in common neurodegenerative disease, but as the authors recognize, larger follow-up studies will be necessary to confirm and document recurrence of brain somatic mutations impacting signaling.

Majors comments

1. The finding of PIN1 mutation in AD-1444 is very interesting. It would be more compelling if the authors could demonstrate that the site of the origin of the tangles in the subject's brain were in regions containing the mutation, compared with non-tangle-containing regions.
2. Page 5, Paragraph 1, no significance differences were observed in terms of VAF and number of variants between AD and non-AD samples. Previous work (PMID 29217584) showed some forms of neurodegeneration are associated with increased abundance of somatic mutations, but these were neurodegenerative conditions that predispose to increased mutations (XP and CS). Have the authors compared the severity of those variants using quantitative prediction tools other than different annotation categories? Is the composition of MAF in the different categories distinct? These will make better use of the VAF estimations.
3. Page 6, Paragraph 1, the authors should also look into the 'mutation signatures' in control brain and blood, to specify whether the differences they observed were related to AD or reflect differences between brain and blood samples. Clonal hematopoietic mutations were demonstrated (PMID 25326804; PMID 30185910), so it is likely that this effect contributes to the increased mosaicism detected in blood, but this should be tested directly.
4. The title 'Brain somatic mutations ... contribute to the initiation of tau pathology in AD', is not supported by the conclusions. The 26.9% of AD individuals (14/52) with at least one pathogenic mutation affecting Tau phosphorylation is intriguing, but the number of controls use for this

comparison is too low for rigorous analysis, and readers will doubt the conclusions. Authors should keep their conclusions as speculative at current, especially given the potential for conflict-of-interest stated in the 'competing interests' section of the paper.

5. There have been extensive studies of brain somatic mosaicism in AD using targeted or exome sequencing, (PMID 30214067, 30114415). Authors should compare their results with existing studies and at least discuss about the differences they each find. Especially recent findings of 'APP gencDNAs' in AD (PMID 30464338), they could analyze their sequencing files to see whether there are potential 'junction reads' or recombination 'gencDNAs' variants in genes like APP.

Minors comments

1. Page 4, Paragraph 1, the optimized cut-off value of Ebscore was calculated from data of 21 healthy control brains and four schizophrenia brains, the assumption is that and the mutation themselves are under a comparable situation, did the author indicate anywhere that the benchmarking sequencing data were generated under comparable pipelines?
2. Page 6, Paragraph 2, for pathogenic variants, do they have different VAFs compared with benign ones?
3. Authors should pay more attentions to the format of the manuscript and check the citations. We think that citation 27 and citation 60 is the same publication. Other examples of duplicated citations includes citation 75 and citation 76, which also means that citation 76 can't be a supporting evidence for citation 75 in Page 10, Paragraph 2, Line 5-7.
4. For the ultra-deep amplicon sequencing validation results provided in Table S4-1, the authors might discuss how validated variants that are filtered out, seen from the results as Mutect only report tissue-specific variants most variants shared by both tissue or not enriched in brain are excluded.
5. According to the existing phenotype info (<https://www.jax.org/strain/025883>) no behavioral studies have been carried out for mPin1 mutant mouse.

Reviewer #2 (Remarks to the Author):

The presented manuscript describes the analysis of somatic variants in brains with AD and normal brains. The study includes (1) sequencing from normal and AD brains and from matching blood; (2) discovering somatic mutations; (3) comparing the spectrum of somatic mutations; (4) identifying enrichment of pathogenic variants in a few pathways related to Tau pathology; (5) testing mutations in PIN1 for underlying phenotype related to AD; (6) estimating the contribution of germline and somatic variants to the etiology of AD.

Overall, this is a novel and interesting study that reports results that will be of interest not only to people working on AD but to the scientific community at large. However, I encountered a few weaknesses and some confusions that need to be addressed.

* Calling of somatic variant, validation and the description of the calling and validation is confusing. Why did the authors use MuTect and then the Bayesian framework (EBscore)? Isn't it enough to use only one approach? Calling across samples is heterogeneous: some variants are called from paired comparison and some are from a single sample.

* Only 29 SNVs were used to calibrate the Bayesian cut-off score. That does not seem enough counts as Supp. Fig. 2 is "bumpy". What were the criteria to consider a call validated? My understanding is that they compared amplicon-seq results for brain and liver. Is that right? Supplementary Table 4 lists validation results, and the conclusions in there are reasonable. The authors say: "If such a mutation was present as statistically reliable compared to previously estimated background errors, we considered using them as true calls." Does it mean that they didn't use a control sample? If not, then this is a potential issue. The error may not be uniform and may be systematic at some sites. The systematic error can be the same in the sequencing

machine, as it also uses PCR. So this may lead to a false call and then to a false validation. What do the authors do with calls for which they can't design primers? I would count them as not validated.

* They say, "We were able to estimate that HIF somatic SNVs increase a rate of 0.53 somatic SNVs per exome per year." It is not clear how to interpret this number. Per cell? Per division? Per what? Because of this confusion I don't think it is worth showing panels A and B in Fig. 2 in the main text. I suggest removing them or moving them to the supplement.

* Not sure why they need to filter out variants by gnomAD before doing enrichment analysis. This seems ad hoc. Pathway enrichment analysis was done with Enrichr, and I couldn't find the description whether that server takes into account the gene length when calculating enrichment. This is a critical point. Longer genes are more likely to be mutated. So whatever enrichments the authors claim can be merely the results of genes being longer. The permutation test that the authors conduct does not account for gene length either.

Smaller points:

* It is not clear what the authors mean by "excluding those with average base positions of either <1 or >100".

* Manual inspection makes the results subjective, and how to interpret the Blat score, namely "the second highest blat score was <900"?

* I think that the authors should refer to somatic mutations as "putative pathogenic" instead of "pathogenic".

* One paragraph in the introduction basically repeats the Abstract. I don't think this paragraph is necessary and propose removing it.

Reviewer #3 (Remarks to the Author):

Park et al. report results on a 2x2 design between Alzheimer's disease cases and controls, and matched brain (hippocampal formation regions) and blood. The results are interesting and the study was well done. I find the mutational profile and PIN1 results particularly interesting, and believe they support important hypotheses found in the Alzheimer's disease field. I have only minor concerns for the authors to address:

1. Authors "examined somatic mutations with low allelic frequency at the genome-wide level".

They should specify and (more importantly) justify the cutoff in the main text.

2. My primary concern is regarding the 15 AD samples lacking matched blood. Authors included 15 samples without matched blood and applied more stringent criteria to identify somatic mutations.

This approach is probably reasonable, but the authors need to show the 15 samples lacking matched blood are not statistically and meaningfully different from samples with matched blood.

i.e., do they have significantly different mutation profile in quantity, accuracy with TASEq (as reported in figure 1b), variant allele frequency (figure 1c)? I also wonder the six outliers in figure 1d in AD brain are from these unmatched samples. Although there is no statistical difference, some readers may conclude there is a 'trend', given the outliers. If there is a significant difference between samples with and without matched blood, the samples without matched blood need to either be excluded, or variant inclusion thresholds need to be modified. This study should as 'clean' as possible.

3. Authors report mutational signatures in Figure 2 c,d. Neither the text nor the figure legend makes it clear which refers to brain and blood. Readers can deduce based on the reported

signatures in the text, but authors should make it clear (e.g., "Fig. 2c" should be at the end of the sentence referring to brain data).

4. I commend authors for performing a permutation test. This helps validate the results are not spurious. Authors should report the p-values for the permutation test (0.0019, 0.0068, and 0.0237) in the main text, rather than making the reader look in the methods.

5. 'abnormally' should be 'abnormal' in 'Previous studies have highlighted abnormally increases in...'.

Reviewer #1

Remarks to the Author:

In this manuscript, Park and colleagues tests the hypothesis that brain somatic mutations arising in the hippocampus contributes to Alzheimer's disease (AD). They applied ~600X coverage whole-exome sequencing (WES) on 111 postmortem hippocampi (HIF) and matched blood samples from 52 AD and 11 control cases. They observed age-related accumulation of variants in laser-capture enriched HIF cells, concluding that an average of 0.53 de novo SNVs/exome occur per year, and 4.8-fold more occur in blood, and extrapolate across the whole genome to conclude that 21.32 de novo SNVs/genome occur per year. Amplicon sequencing showed validation in 80.3% of variants tested, suggesting that false-discovery was minimal. Previous work (PMID 29217584) identified ~40 de novo SNVs/year in the hippocampal dentate gyrus, which is remarkably similar to Park's results, given that different methods were used to arrive at these conclusions. Mutational signatures were also similar to the previous work,

pointing to aging, DNA damage, and reactive oxygen species. There were no differences in mutation counts nor genomic location of the somatic mutations in AD vs. non-AD tissues, but AD brains showed enrichment for mutations in signaling pathways, occurring in 26.9% of AD individuals. This was one of the most exciting findings of the paper but the authors do not probe it further. The authors found a mutation in the proline isomerase PIN1, in patient AD-1444, demonstrate that it is a destabilizing point mutation, and functionally demonstrate that shRNA knockdown of Pin1 leads to increased Tau phosphorylation and oligomerization.

The methodology for identifying and validation somatic variants are reliable, and statistical analysis and benchmark are well performed. The manuscript is well-written and easy to understand. Compared with existing targeted sequencing based results, the manuscript provides a comprehensive assessment of somatic mutations and confirmed previous predictions. The manuscript sheds light on the potential for somatic mutations in common neurodegenerative disease, but as the authors recognize, larger follow-up studies will be necessary to confirm and document recurrence of brain somatic mutations impacting signaling.

Response:

We thank R1 for the positive comments on our manuscript. We addressed all concerns below.

MAJOR QUESTIONS

Q1. *The finding of PIN1 mutation in AD-1444 is very interesting. It would be more compelling if the authors could demonstrate that the site of the origin of the tangles in the subject's brain were in regions containing the mutation, compared with non-tangle-containing regions.*

Response:

We totally agree with R1's suggestion and thus performed laser capture microdissection (LCM) followed by targeted amplicon sequencing on neurofibrillary tangle-containing neurons in the entorhinal cortex, compared to AT8-negative neurons. We used commercially available monoclonal AT8 antibody (MN1020, Thermo) to label hyperphosphorylated Tau proteins in neurons of the entorhinal cortex from patient AD-1444 with PIN1 c.477 C>T (p.Thr152Met). Consistent with our hypothesis, the VAF of the mutation, which was 1.8% in the bulk HIF tissue, was significantly enriched by 4.9-fold (VAF=8.75%) in AT8-positive neurons (n=56), whereas this mutation was not detected in AT8-negative neurons (n=52) in the entorhinal cortex (**Supplementary Fig. 6, Table 10, and Methods**). This result suggests that entorhinal cortical neurons carrying the pathogenic somatic mutation in PIN1 are likely to be the site of origin of the Tau pathology. In the revised manuscript, we added this result and related comments on p 7-8, Lines 197-204.

Q2. *Page 5, Paragraph 1, no significance differences were observed in terms of VAF and number of variants between AD and non-AD samples. Previous work (PMID 29217584) showed some forms of neurodegeneration are associated with increased abundance of somatic mutations, but these were neurodegenerative conditions that predispose to increased mutations (XP and CS). Have the authors*

compared the severity of those variants using quantitative prediction tools other than different annotation categories? Is the composition of MAF in the different categories distinct? These will make better use of the VAF estimations.

Response:

In the original manuscript, we utilized Phred-scaled CADD score¹ for quantitative prediction of the severity of post-filtered variants and considered variations with a CADD score >20 as putatively pathogenic variants. Multiple algorithms have been developed for predicting the degree of deleteriousness based on evolutionary conservation, sequence homology, protein structure, and etc. Although individual methods are often useful, they are not directly comparable to each other since they utilize the different information for the prediction. Also, there is no definite gold standard method for such prioritization of the variant. We decided to use the latest version of CADD GRCh38-v1.4 as it systemically annotates both coding and non-coding variants and has the fewest missing values for variants compared to other dbNSFP listed tools (**Fig. R1a and Table R1**). There was no significant difference in the average pathogenicity scores (e.g., CADD, MetaLR, M-CAP, Polyphen2, SIFT, MutPred) of brain or blood somatic mutations between AD and non-AD groups (**Fig. R1b**). We also examined the composition of variant allele fractions (VAF) of different annotation categories as R1 suggested (**Fig. R1c**). In 622 post-filtered brain somatic mutations of Alzheimer's disease, the VAFs of non-synonymous mutations were not significantly different from those of synonymous, UTR, splicing, and intergenic variants and slightly lower than those of intronic mutations ($P < 0.0001$). Interestingly, putative pathogenic variants in AD brains showed significantly lower VAF ($P = 0.0004$), compared to that of benign variants (**Fig. R1d**). Such differences in VAF between pathogenic and benign variants were also observed in non-AD brains ($P = 0.001$), non-AD blood ($P = 0.0228$), and AD blood ($P = 0.0002$) as well. This result implies that pathogenic somatic mutations are likely to be found in a smaller fraction of cells, perhaps due to their deleterious effects on the viability, proliferation, or clonal expansion of mutation-carrying cells.

a

Method	Missing values for SNVs (%)	Threshold	Description
CADD (v1.4)	4.02 (25/622)	Phred score > 20	63 distinct variant annotation retrieved from Ensembl Variant Effect Predictor (VEP), ENCODE project, and UCSC genome browser
MetaLR	63.67 (396/622)	score > 0.5	Logistic regression to integrate 9 deleteriousness prediction scores and maximum minor allele frequency for more accurate and comprehensive evaluation of deleteriousness of missense mutations
M-CAP	64.15 (399/622)	score > 0.025	Combining amino acid conservation features with gradient boosting trees and computing scores trained on mutations linked to Mendelian diseases
Polyphen2_HDIV	64.63 (402/622)	score > 0.5	Merging 8 protein sequence features and three protein structure features
SIFT	64.95 (404/622)	(1-score) > 0.95	Protein sequence conservation among homologs
MutPred	69.94 (435/622)	score > 0.5	Protein sequence-based model using SIFT and a gain/ loss of 14 different structural and functional properties

b**c****d**
Fig. R1: Comparison of the pathogenicity and variant allele frequency of brain and blood somatic mutations in Alzheimer's disease and non-demented control cases.

a, Summary of pathogenicity prediction methods analyzed in our study. Missing values for each method are calculated. **b**, Violin plots displaying CADD, Polyphen2_HDIV, SIFT, and MutPred score of post-filtered brain and blood somatic mutations in AD and non-AD groups. **c**, Comparison of variant allele fractions of AD brain somatic mutations in six different functional categories. **d**, Comparison of variant allele fractions of CADD categorized benign and pathogenic variants of brain and blood somatic mutation in AD and non-AD. *P* values were calculated by one-way ANOVA test, followed by post hoc multiple comparison in **b** and **c**. *P* values were calculated by unpaired t-tests accompanied with Welch's correction in **d**. Abbreviations: n.s., not significant; *, $p < 0.05$; ***, $p < 0.001$; ****, $p < 0.0001$.

Q3. Page 6, Paragraph 1, the authors should also look into the 'mutation signatures' in control brain and blood, to specify whether the differences they observed were related to AD or reflect differences between brain and blood samples. Clonal hematopoietic mutations were demonstrated (PMID 25326804; PMID 30185910), so it is likely that this effect contributes to the increased mosaicism detected in blood, but this should be tested directly.

Response:

Fig. R2: Mutation signatures of brain and blood somatic mutations in non-AD.

The best decomposed mutation signature models from multiple likelihood estimation were derived for each tissue along with actual distribution of 96 possible mutation types. **a**, Single base substitution (SBS) signatures 4, 1, and others account for 37.3%, 16.2%, and 46.5% of brain somatic mutations in non-AD, respectively. **b**, SBS signatures 5 and 1 account for 74.2%, and 25.8% of blood somatic mutations in non-AD, respectively. **c**, Merging brain somatic mutations from both AD and non-AD samples show SBS signatures 1, 18, and others which account for 18.5%, 24.5%, and 57.0%, respectively.

We thank R1 for this constructive comment. Since the number of non-AD individuals (11 cases) was small, we were able to utilize only 132 of somatic variants in non-AD brains and 355 in non-AD blood for mutation signature analyses compared to those in 52 cases of AD brains (595 SNVs) and AD blood (2475 SNVs). Thus, it could be inaccurate to infer mutation signatures in such small number of somatic mutations, especially in non-AD brains. Nevertheless, we derive mutation signature analyses in both non-AD brain and non-AD blood. In non-AD blood, SBS signatures 5 and 1 can explain 74.2% and 25.8% of all somatic mutations, consistent with the finding that in AD blood (**Fig. R2b, Fig. 2, and Table R2**). However, we observed that SBS signatures 4 and 1 accounted for 37.3% and 16.2% of all somatic mutations in non-AD brains (**Fig. R2a**). Unlike mutation signature in AD brains, the mutation signature of DNA damage by reactive oxygen species (SBS signature 18) was not detected in non-AD brains, possibly due to much smaller number of analyzed somatic mutations. On the other hand, when we pooled brain somatic mutations found in both AD and non-AD altogether, we were still able to observe SBS signature 18 as well as SBS signature 1 (**Fig. R2c**).

Also, we agreed with R1's suggestion for analyzing clonal hematopoietic mutations in blood samples. Mature blood and immune cells are produced by the process of hematopoiesis. As people age, clonal expansions of mutated stem cells within blood frequently occur, which is associated with increased numbers of somatic mutations and an increased risk of developing hematological malignancies subsequent thereto². We found that the VAF and number of somatic mutations in the AD blood were significantly higher than those in the AD brain (**Figs. 1d,f**). As R1 mentioned, this finding may indicate that the clonal hematopoiesis-mediated subpopulation of cells is more diverse in blood than that in brain tissue. To test the clonal hematopoiesis in blood, we adopted a clonal reconstruction algorithm that utilizes variant allele frequency (VAF) of somatic single nucleotide variants (SNV) obtained from next-generation sequencing data to decompose clonal composition in given samples. Currently available methods, including LICHEeE algorithm³, use the VAF to cluster distinct single nucleotide variations and build statistical constraint model to infer the clonal evolution⁴. We performed VAF-based SNV clustering and subsequent clonal lineage reconstruction according to the author's guideline with modification of the following parameters to account for delineating clusters with low-level somatic mutations: (*-maxVAFAbsent 0.005 -minVAFPresent 0.005 -minClusterSize 1 -maxClusterDist 0.005*). As a result, we found out that blood cells have significantly a higher number and distinctive pattern of sub-clones than those of brain cells (**Supplementary Fig. 11**). Together with mutation signatures of somatic mutations in blood, this result suggests that clonal hematopoiesis contributes to the increased number and VAFs of somatic mutations in patient blood. We added this result and related comments on p 10, Lines 265-272 in the revised manuscript.

Q4. *The title 'Brain somatic mutations ... contribute to the initiation of tau pathology in AD', is not supported by the conclusions. The 26.9% of AD individuals (14/52) with at least one pathogenic mutation affecting Tau phosphorylation is intriguing, but the number of controls use for this comparison is too low for rigorous analysis, and readers will doubt the conclusions. Authors should keep their conclusions as speculative at current, especially given the potential for conflict-of-interest stated in the 'competing interests' section of the paper.*

Response:

We thank R1 for the constructive criticism on the title of the original manuscript. We agree that our findings on a limited cohort are speculative at this time. Therefore, we changed the original title to "Brain somatic mutations associated with aging contribute to dysregulation of Tau phosphorylation in Alzheimer's disease" in the revised manuscript.

Q5. *There have been extensive studies of brain somatic mosaicism in AD using targeted or exome*

sequencing, (PMID 30214067, 30114415). Authors should compare their results with existing studies and at least discuss about the differences they each find. Especially recent findings of 'APP gencDNAs' in AD (PMID 30464338), they could analyze their sequencing files to see whether there are potential 'junction reads' or recombination 'gencDNAs' variants in genes like APP.

Response:

Wei et. al⁵ [PMID 30214067] reported 56 brain somatic mutations in 1461 human brain exomes (mean depth 51.9X) including 277 Alzheimer's disease cases. Their pipeline utilized GATK haplotype caller (v3.4) to call heterozygous variants along with filtering out by variant allele fraction (e.g., 10% < filtered in VAF < 50%) and subsequent exclusion of common variants (e.g., ExAC Minor Allele Frequency > 5%). Among 56 post-filtered variants from their pipeline, 22 (55%, 22/40) of the variants were further validated by targeted amplicon sequencing (Miseq) or Pyrosequencing (PyroMark Q24). With 22 validated somatic mutations in 15 different patients, they examined mutation signature (COSMIC Signature 6 - defective DNA mismatch repair) and extrapolated mutation rate in human brains (4.25×10^{-10} per base pair per individual).

Meanwhile, Nicolas et. al⁶ [PMID 30114415] performed molecular barcode panel sequencing on 11 known autosomal dominant and risk factor genes in Alzheimer's disease by using smMIP probes (single-molecule molecular inversion probes) on 335 blood and 100 brain samples from AD patients. After filtering out variants by base-specific error rate and PCR duplicates, they identified nine somatic variants (*APP*, *MARK4*, *NCSTN*, *SORL1*; 0.2% < VAF < 10.8%) in the smMIP libraries (mean depth 2576X) and subsequently validated those variants in nine different patients with targeted amplicon sequencing (Ion Torrent PGM).

Although we performed deep sequencing in 52 AD brains at the genome-wide level, followed by more specific bioinformatic analysis of low-level somatic mutations, we could not detect any of the 22 somatic mutations from Wei et. al or the nine variants from Nicolas et. al. Also, given that our pipeline may miss these variants because of different algorithmic approaches, we independently investigated those altered alleles from analysis-ready bam files using the bam-readcount R package (**Supplementary Table 13 and Methods**). Neither brain nor blood samples from the 52 AD and 11 non-AD cases enrolled in our study carried more than three reads supporting AD-associated variants (e.g., 2/22 variants in Wei et. al, 9/9 variants in Nicolas et. al) mentioned in these two studies. Meanwhile, we found a variant in *LAMA5* from 1 FTD-ALS patient (rs753153265) was observed in HIF of AD-196 with 53.6% of variant allele frequency and further verified regarding heterozygous germline SNP through GATK HaplotypeCaller (VAF=51.98%).

Lee et. al⁷ [PMID 30464338] reported that somatic *APP* gene recombination from Alzheimer's disease generates thousands of previously unknown gene variants characterized as genomic complementary DNAs (gencDNAs), which could show identical sequences to cDNAs copied from brain-specific spliced RNAs, as well as a myriad of truncated forms characterized by exonic deletions and intra-exonic junctions (IEJs) to produce novel sequences that become retro-inserted into the genome of single neurons. Such recombination in *APP* gene appeared to require gene transcription, reverse transcriptase activity, and DNA strand breaks. According to Lee et. al, both the numbers and the patterns of *APP* gencDNAs were altered and increased in sporadic Alzheimer's disease. Although our deep whole exome sequencing data (c.f. LCM-enriched neurons from hippocampal formation; whole exome library; 101bp paired-end [Hiseq2500]) is not exactly parallel to what Lee et. al used to infer the presence of IEJs in the *APP* gene (c.f. NeuN-positive frontal cortical neurons; Targeted *APP* gene-captured library; 151bp paired-end [Nextseq]), we analyzed our raw fastq files according to the author's guideline to examine the presence of IEJs.

Briefly, raw fastq files were aligned to the human reference genome (GRCh38) using STAR aligner. Duplicate reads were marked and removed by Picard and Bam files are then converted to bed12 format to be visualized in exonjunction R package. As a result, we were able to observe 72 IEJs and six two-exon-exon junctions in the sequencing data (SRX4741695) from three sporadic AD patients which are mentioned in the paper. Meanwhile, we could detect 1~17 IEJs from four different AD patients. No IEJs, however, were detected in 11 non-demented control cases. We added these results on p 11, Lines 287-292 in the revised manuscript.

MINOR QUESTIONS

Q6. Page 4, Paragraph 1, the optimized cut-off value of Ebscore was calculated from data of 21 healthy control brains and four schizophrenia brains, the assumption is that and the mutation themselves are under a comparable situation, did the author indicate anywhere that the

benchmarking sequencing data were generated under comparable pipelines?

Response:

We thank R1 for pointing out the lack of detailed information on the panel of 21 normal and four test datasets in the manuscript. As we previously mentioned in **Supplementary Table 2**, 25 matched brain and liver samples were produced under the same pipeline as AD and non-AD cohort. Briefly, we used Agilent V5+UTR exome library prep kit and Illumina HiSeq2500 sequencer to ensure >500X average sequencing depth and subsequent running of the MuTect algorithm to find somatic mutations. Along with additional calibration for EBscore (R2's comment), we updated information on the exome library kit, sequencer type, and average sequencing depth on related columns of **Supplementary Table 2** and **methods** section in the revised manuscript.

Q7. *Page 6, Paragraph 2, for pathogenic variants, do they have different VAFs compared with benign ones?*

Response:

As we mentioned in "**Response to Q2**", putative pathogenic variants show significantly lower VAFs than benign variants regardless of disease status or origin of tissue.

Q8. *Authors should pay more attentions to the format of the manuscript and check the citations. We think that citation 27 and citation 60 is the same publication. Other examples of duplicated citations includes citation 75 and citation 76, which also means that citation 76 can't be a supporting evidence for citation 75 in Page 10, Paragraph 2, Line 5-7.*

Response:

We apologize for duplication and unmatched references in the previous manuscript. We removed all the duplicates and updated the reference list in the revised manuscript (Page 11; Lines 298-299).

Q9. *For the ultra-deep amplicon sequencing validation results provided in Table S4-1, the authors might discuss how validated variants that are filtered out, seen from the results as Mutect only report tissue-specific variants most variants shared by both tissue or not enriched in brain are excluded.*

Response:

The application of targeted amplicon sequencing on 159 brain somatic mutations allowed us to identify 77 true somatic mutations (**Supplementary Table 4**). Among these 77 variants, 67 were correctly identified by MuTect+Post-call filters, while 10 were false negatives, defined as true somatic mutations incorrectly filtered out by the Post-call filter processes due to low EBscore (≤ 2.396), high average second highest BLAT score (>900), or both. As we tried to systemically rule out genome-wide sequencing or alignment artefacts from deep whole exome sequencing data ($>500X$), we adopted quantitative filters (e.g., coverage depth, variant allele frequency) along with EBscore⁸ and visual inspection^{9,10}. Considering an ideal post-call filter would increase precision to 100% without any decrease in sensitivity, we are aware of the limitation of these filters in securing all true calls. Nevertheless, we could increase precision rate from 48.4% to 79.8% by missing only 10 variants out of 77 true calls using the aforementioned post-call filters (**Fig. R3d in response to R2-Q1**).

Q10. *According to the existing phenotype info (<https://www.jax.org/strain/025883>) no behavioral studies have been carried out for mPin1 mutant mouse.*

Response:

We thank R1 for mentioning currently available murine Pin1 knock-out mouse line. According to Liou et. al¹¹, where they developed the whole-body knock out of Pin1, this mice showed age-dependent motor and behavioral deficiencies as did human Tau harboring transgenic mouse: retinal atrophy, abnormal limbic clasping reflexes, hunched posture, reduced mobility, and eye irritation. Interestingly, they showed age-dependent neuropathy, hyperphosphorylation of Tau, and NFT-like tangles in the brain. Therefore, our finding in loss-of-function *PIN1* mutation (c.477 C>T) and subsequent functional validation are compatible with the *in vivo* data of Liou et. al¹¹.

Reviewer #2

Remarks to the Author:

The presented manuscript describes the analysis of somatic variants in brains with AD and normal brains. The study includes (1) sequencing from normal and AD brains and from matching blood; (2) discovering somatic mutations; (3) comparing the spectrum of somatic mutations; (4) identifying enrichment of pathogenic variants in a few pathways related to Tau pathology; (5) testing mutations in PIN1 for underlying phenotype related to AD; (6) estimating the contribution of germline and somatic variants to the etiology of AD.

Overall, this is a novel and interesting study that reports results that will be of interest not only to people working on AD but to the scientific community at large. However, I encountered a few weaknesses and some confusions that need to be addressed.

Response:

We thank R2 for the positive comments on our manuscript. We addressed all concerns below.

MAJOR QUESTIONS

Q1. *Calling of somatic variant, validation and the description of the calling and validation is confusing. Why did the authors use MuTect and then the Bayesian framework (EBscore)? Isn't it enough to use only one approach? Calling across samples is heterogeneous: some variants are called from paired comparison and some are from a single sample.*

Response:

We apologize for confusing the description of our somatic mutation calling pipeline. Firstly, in case of the 48 cases from the Netherlands Brain Bank, we could acquire and sequence both brain and blood tissues, thereby using paired sample mode of MuTect v1.7 for calling both brain and blood somatic mutation. Meanwhile, in the 15 cases where matched peripheral tissues were not available, brain somatic mutations were analyzed using the single sample mode of MuTect v1.7 along with more strict quantitative filters (e.g., Depth \geq 100; VAF $<$ 20%) than that of the 48 matched tissues available cases (e.g., Depth \geq 35; VAF $<$ 40%) (**Fig. R3b**). Nonetheless, as seen in **Supplementary Fig. 4** and **Table 4**, post-filtered somatic mutations calls from unmatched samples also showed comparable precision from targeted amplicon sequencing (TASeq) result.

Although the MuTect algorithm is known to sensitively and accurately detect somatic mutations in tumors, we are aware of that its performance is not good enough for accurately detecting low-level somatic mutations from deep sequencing data due to the high false positive rate^{12,13}. Therefore, we applied the Empirical Bayesian Filter (EBscore), which rules out error-prone sites and sequencing errors⁸ using a panel of normals, on raw MuTect calls along with quantitative and qualitative filters to increase the precision of calling somatic mutation. Applying all three post-call filters on raw MuTect calls increased the precision rate ($=\frac{\text{True Positive}}{\text{True Positive}+\text{False Positive}}$) in both the independent 54 variants from 11 Schizophrenia brain samples (Test dataset) (**Fig. R3c**) and the 159 variants from 55 Alzheimer and non-demented control brain samples (**Fig. R3d**).

Fig. R3: Evaluation of EBscore cutoff and Post-call filter processes.

a, Two receiver operator characteristic (ROC) curves were drawn to determine the cut-off value of EBscore using targeted amplicon sequencing results from randomly picked 29 and 54 brain somatic mutations from four and eleven schizophrenia brains. Incrementing test-data set up to 54 variants showed same cutoff value of EBscore. **b**, Detailed schematics of bioinformatic analysis pipeline used for detecting somatic SNVs from matched and unmatched samples. Specific conditions for unmatched cases are denoted in red. **c**, Comparison of precision rate of raw MuTect calls and independently applied post-call filters using 54 variants from Schizophrenia brains. **d**, Comparison of precision rate of raw MuTect calls and independently applied post-call filters using 159 variants from Alzheimer's disease and non-demented control brains.

Q2. Only 29 SNVs were used to calibrate the Bayesian cut-off score. That does not seem enough counts as Supp. Fig. 2 is "bumpy". What were the criteria to consider a call validated? My understanding is that they compared amplicon-seq results for brain and liver. Is that right?

Supplementary Table 4 lists validation results, and the conclusions in there are reasonable. The authors say: "If such a mutation was present as statistically reliable compared to previously estimated background errors, we considered using them as true calls." Does it mean that they didn't use a control sample? If not, then this is a potential issue. The error may not be uniform and may be systematic at some sites. The systematic error can be the same in the sequencing machine, as it also uses PCR. So this may lead to a false call and then to a false validation. What do the authors do with calls for which they can't design primers? I would count them as not validated.

Response:

To set the proper threshold of EBscore on raw MuTect calls, we initially sequenced four Schizophrenia cases and 21 non-schizophrenia cases, called somatic SNVs using the same pipeline, and used them as a panel of normals (**Supplementary Fig. 2 and Table 3**). As we agree to R2's concern about the possible shortage of data for calibrating the EBscore, we increase the test dataset to 54 variants from 11 Schizophrenia patients. Although the area under curve for EBscore was slightly lower than that of previous observation (**Fig. R3a**), the cutoff value to maximize 'sensitivity + specificity' was still "2.396," suggesting that our initial threshold was properly determined.

To test whether post-filtered calls are true calls, as seen in **Supplementary Table 4**, we randomly selected 11 % (84/764 SNVs) of filtered somatic SNVs from the HIF specimens and performed targeted amplicon sequencing (5,434X to 4,797,498X of read-depth) on "both" brain and blood samples. For interpreting the results of targeted amplicon sequencing, we considered variants as validated when their variant allele fraction in TASEq was significantly higher than the base-specific background error rate that we recently reported for amplicon-based Illumina platforms in control samples: T>A (0.00312107), T>C (0.007970457), T>G (7.58E-04), C>T (0.004071926), C>G (7.65E-04), C>A (0.001847634)¹².

Specific primer sequences were able to be designed for all variants-of-interest (141/141 for matched and 18/18 variants for unmatched samples). Detailed primer sequence information is listed in **Supplementary Table 4-2**.

Q3. They say, "We were able to estimate that HIF somatic SNVs increase a rate of 0.53 somatic SNVs per exome per year." It is not clear how to interpret this number. Per cell? Per division? Per what? Because of this confusion I don't think it is worth showing panels A and B in Fig. 2 in the main text. I suggest removing them or moving them to the supplement.

Response:

We apologize for confusing with the rate of somatic SNVs. Since we enriched neuronal cells of HIFs by performing laser capture microdissection and detected somatic SNVs with a VAF of at least 0.52%, we rephrased the statement as "somatic SNVs with VAF of at least 0.52% in neuronal cells of HIF increase a rate of 0.53 somatic SNVs per exome per year". As R2 suggested, we also moved these figures to the supplementary figures (**Supplementary Figs. 10a,b**).

Q4. Not sure why they need to filter out variants by gnomAD before doing enrichment analysis. This seems ad hoc. Pathway enrichment analysis was done with Enrichr, and I couldn't find the description whether that server takes into account the gene length when calculating enrichment. This is a critical point. Longer genes are more likely to be mutated. So whatever enrichments the authors claim can be merely the results of genes being longer. The permutation test that the authors conduct does not account for gene length either.

Response:

We thank R2 for the constructive comments. Since rare variants are more likely to be pathogenic or relevant to the disease, we excluded variants with $MAF \geq 0.01\%$ in gnomAD for our enrichment analysis based on recent studies with rare variants^{14,15}.

We agree with R2's concern about the gene length. As R2 pointed out, the Enrichr algorithm¹⁶ does not innately account for gene length in the analysis. To exclude such gene length bias, we independently performed gene length adjusted gene-set enrichment test with DNENRICH software¹⁷. Briefly, we separately all collected genes with putatively pathogenic somatic mutations in brain and blood tissues from Alzheimer's disease patients and non-demented controls (AD_Brain, AD_Blood, non-AD_Brain, non-AD_Blood). Then, we performed gene-set enrichment analysis by DNENRICH with 100,000 permutations to identify critical biological processes in which pathogenic variants were overrepresented. The KEGG database (v.2016) was used as a reference database to find AD-relevant

pathways. Consistent with our previous result with the Enrichr algorithm, AMPK, MAPK, and PI3K-AKT pathways were still significantly enriched only in AD brain samples (**Supplementary Figs. 5b,c**). Therefore, our results suggest that putatively pathogenic somatic mutations in HIF of AD individuals are significantly enriched for the PI3K-AKT, MAPK, and AMPK pathways. We added this result and comments in the revised manuscript on p 7, Lines 177-180.

MINOR QUESTIONS

Q5. *It is not clear what the authors mean by “excluding those with average base positions of either <1 or >100”.*

Response:

We apologize for the unclear description on filtering conditions in our pipeline. As the base quality and sequencing accuracy drop at the end of the reads from Illumina platform¹⁸, we considered altered alleles positioned at the end of each read (base position=0 or 101 for 101-bp paired end reads) as unreliable calls. To clarify such filtering process, we modified the description of the process as “excluding variants with all supporting reads located at either end of reads” in the revised manuscript.

Q6. *Manual inspection makes the results subjective, and how to interpret the Blat score, namely “the second highest blat score was <900”?*

Response:

Manual inspection with the IGV viewer on aligned reads of variant calls is useful and frequently used to remove ambiguous mapping and sequencing errors⁹. By performing a BLAT search embedded in the IGV viewer, one can ensure whether the read of interest is uniquely mapped on the reference genome. The average of the highest BLAT score of genomic region around the variant-of-interest is always >950 for 101-bp paired end reads from an Illumina sequencer (**Supplementary Table 3 and Fig. R4**). Although there is yet no clear cut value for demarcating ambiguously mapped reads in the field¹⁰, based on our targeted amplicon sequencing results, we set the average second highest BLAT score <900 as a filtering condition for uniquely mapped reads.

a

Subject ID: SCZ_1001
Gene: GMPPB
Mutation: chr3:49727580 G>A
Top Blat score: 980-980 (980)
2nd Highest Blat score: 198-267 (237)

**deep
WESeq**

**Liver
PFC**

TASeq

**Liver
PFC**

b

Subject ID: SCZ_1002
Gene: RBMX
Mutation: chrX:136877960 C>T
Top Blat score: 930-950 (942)
2nd Highest Blat score: 900-930 (925)

**deep
WESeq**

**Liver
PFC**

TASeq

**Liver
PFC**

Fig. R4: IGV browser images of true and false call from from test dataset.

a, True calls show one mismatched base pair in each read unless its homozygous/heterozygous SNPs. The average of the highest blast score for supporting reads with altered alleles was always >950 and the average of 2nd highest blast score is <900. **b**, False calls show multiple mismatched base pairs (e.g., insertion) in supporting reads. The average of the second highest blast score for supporting reads was ≥ 900 .

Q7. I think that the authors should refer to somatic mutations as “putative pathogenic” instead of “pathogenic”.

Response:

We appreciate R2 for suggesting the better expression. In the revised manuscript, we referred to somatic mutations as “putatively pathogenic” instead of “pathogenic.”

Q8. One paragraph in the introduction basically repeats the Abstract. I don’t think this paragraph is necessary and propose removing it.

Response:

We thank R2 for pointing out the redundant description in the introduction. As R2 suggested, we removed that paragraph on p3 in the revised manuscript.

Reviewer #3

Remarks to the Author:

Park et al. report results on a 2x2 design between Alzheimer's disease cases and controls, and matched brain (hippocampal formation regions) and blood. The results are interesting and the study was well done. I find the mutational profile and PIN1 results particularly interesting, and believe they support important hypotheses found in the Alzheimer's disease field. I have only minor concerns for the authors to address:

Response:

We thank R3 for the positive comments on our manuscript. We addressed all concerns below.

MINOR QUESTIONS

Q1. *Authors "examined somatic mutations with low allelic frequency at the genome-wide level". They should specify and (more importantly) justify the cutoff in the main text.*

Response:

The cutoff value of variant allele fraction (VAF) in our deep whole exome sequencing data was 0.52%, as seen in **Supplementary Table 4**. The VAF of somatic mutations validated through the targeted amplicon sequencing ranged from 0.52% to 15.3%. We added this on p4 in the revised manuscript.

Q2. *My primary concern is regarding the 15 AD samples lacking matched blood. Authors included 15 samples without matched blood and applied more stringent criteria to identify somatic mutations. This approach is probably reasonable, but the authors need to show the 15 samples lacking matched blood are not statistically and meaningfully different from samples with matched blood. i.e., do they have significantly different mutation profile in quantity, accuracy with TASEq (as reported in figure 1b), variant allele frequency (figure 1c)? I also wonder the six outliers in figure 1d in AD brain are from these unmatched samples. Although there is no statistical difference, some readers may conclude there is a 'trend', given the outliers. If there is a significant difference between samples with and without matched blood, the samples without matched blood need to either be excluded, or variant inclusion thresholds need to be modified. This study should as 'clean' as possible.*

Response:

We agreed with R3's concern on 15 cases without matched peripheral tissue. Therefore, we additionally validated randomly chosen ~10% of variants (18/171) from post-filtered calls from those unmatched samples. As seen in **Supplementary Fig. 4a**, the precision (accuracy) of post-filtered calls from unmatched samples (72.2%) were comparable to that of post-filtered call from matched samples (80.3%) (Fisher's exact T-test = 0.5078). Moreover, strong correlation ($R^2 = 0.81$) was observed when comparing VAFs across individual sequencing platforms (**Supplementary Fig. 4b**). Finally, there was no significant difference in the average number or mean VAF of brain somatic mutations between AD and non-AD groups with or without peripheral tissues (**Supplementary Figs. 4c,d**). Therefore, this result suggests that our strict quantitative filters (e.g., read depth \geq 100, EBscore $>$ 2.396, VAF $<$ 20%) applied on unmatched samples is comparable to the analysis of the matched samples.

Q3. *Authors report mutational signatures in Figure 2 c,d. Neither the text nor the figure legend makes it clear which refers to brain and blood. Readers can deduce based on the reported signatures in the text, but authors should make it clear (e.g., "Fig. 2c" should be at the end of the sentence referring to brain data).*

Response:

We thank R3 for pointing out the unclear annotation of figure 2 c,d. In the revised manuscript, we added tissue information on each **Figs 2a,b** in the legend.

Q4. *I commend authors for performing a permutation test. This helps validate the results are not spurious. Authors should report the p-values for the permutation test (0.0019, 0.0068, and 0.0237) in the main text, rather than making the reader look in the methods (Page 7; Line 169-172).*

Response:

We appreciate R3 for the positive comment about our permutation test on gene-set enrichment test. As R3 pointed out, we corrected the P values of both original permutation test and gene-length adjusted result (R2's comment) in the revised manuscript.

Q5. *'abnormally' should be 'abnormal' in 'Previous studies have highlighted abnormally increases in...'*

Response:

We corrected it in the revised manuscript.

Table legends

Table R1: List of six pathogenicity scores of brain and blood somatic mutations in Alzheimer's disease and non-demented controls

Table R2: Detailed summary report of mutation signature analyses of non-AD brain, non-AD blood, and AD+non-AD brain from Mutalisk tool

References

1. Rentzsch, P., Kircher, M., Witten, D., Cooper, G.M. & Shendure, J. CADD: predicting the deleteriousness of variants throughout the human genome. *Nucleic Acids Research* **47**, D886-D894 (2018).
2. Xie, M. *et al.* Age-related mutations associated with clonal hematopoietic expansion and malignancies. *Nature Medicine* **20**, 1472 (2014).
3. Popic, V. *et al.* Fast and scalable inference of multi-sample cancer lineages. *Genome biology* **16**, 91-91 (2015).
4. Kuipers, J., Jahn, K. & Beerenwinkel, N. Advances in understanding tumour evolution through single-cell sequencing. *Biochimica et biophysica acta. Reviews on cancer* **1867**, 127-138 (2017).
5. Wei, W. *et al.* Frequency and signature of somatic variants in 1461 human brain exomes. *Genetics in Medicine* (2018).
6. Nicolas, G. *et al.* Somatic variants in autosomal dominant genes are a rare cause of sporadic Alzheimer's disease. *Alzheimer's & Dementia* **14**, 1632-1639 (2018).
7. Lee, M.-H. *et al.* Somatic APP gene recombination in Alzheimer's disease and normal neurons. *Nature* **563**, 639-645 (2018).
8. Shiraishi, Y. *et al.* An empirical Bayesian framework for somatic mutation detection from cancer genome sequencing data. *Nucleic Acids Research* **41**, e89-e89 (2013).
9. Robinson, J.T., Thorvaldsdóttir, H., Wenger, A.M., Zehir, A. & Mesirov, J.P. Variant Review with the Integrative Genomics Viewer. *Cancer Research* **77**, e31-e34 (2017).
10. Nishioka, M. *et al.* Identification of somatic mutations in postmortem human brains by whole genome sequencing and their implications for psychiatric disorders: Somatic mutations in the human brain, (2017).
11. Liou, Y.-C. *et al.* Role of the prolyl isomerase Pin1 in protecting against age-dependent neurodegeneration. *Nature* **424**, 556 (2003).
12. Kim, J. *et al.* The use of technical replication for detection of low-level somatic mutations in next-generation sequencing. *Nature Communications* **10**, 1047 (2019).
13. Krøigård, A.B., Thomassen, M., Lænkholm, A.-V., Kruse, T.A. & Larsen, M.J. Evaluation of Nine Somatic Variant Callers for Detection of Somatic Mutations in Exome and Targeted Deep Sequencing Data. *PloS one* **11**, e0151664-e0151664 (2016).
14. Raghavan, N.S. *et al.* Whole-exome sequencing in 20,197 persons for rare variants in Alzheimer's disease. *Annals of clinical and translational neurology* **5**, 832-842 (2018).
15. Mazzarotto, F. *et al.* Defining the diagnostic effectiveness of genes for inclusion in panels: the experience of two decades of genetic testing for hypertrophic cardiomyopathy at a single center. *Genetics in Medicine* **21**, 284-292 (2019).
16. Kuleshov, M.V. *et al.* Enrichr: a comprehensive gene set enrichment analysis web server

- 2016 update. *Nucleic Acids Research* **44**, W90-W97 (2016).
17. Fromer, M. *et al.* De novo mutations in schizophrenia implicate synaptic networks. *Nature* **506**, 179 (2014).
 18. Fuller, C.W. *et al.* The challenges of sequencing by synthesis. *Nature Biotechnology* **27**, 1013 (2009).

Reviewers' comments:

Reviewer #1 (Remarks to the Author):

The authors have been extraordinarily responsive to my multiple questions, and have performed a series of additional control experiments to address my questions, all of which support their model. Furthermore, they expanded the data to include consideration of mutational signatures, clonal hematopoiesis and recent publications in the field of mosaicism in Alzheimer's to make the paper stronger.

Reviewer #2 (Remarks to the Author):

I just have two minor comments

* Manual curation and BLAT results

I don't feel that the SNV chrX:136,877,960 C>T displayed in Figure R4b demonstrates the point. The panel displays two supporting reads for T in WES in liver and 6 supporting reads in WGS for brain. Why MuTect even called it? Isn't is suspicious? The SNV and indel next to it are always co-occurring and both of them have rsIDs (rs76812369 and rs755459969), which suggests that they may be germline. In this regard I think the authors should check and report how many of their putative mosaic variants are found in gnomAD.

Operating by BLAT (or BLAST as the authors write in the caption to figure R4b) score values gives a reader no sense of how reasonable applied cut off are. The author should provide rough correspondence of cut off value to % of sequence identity.

* I'm still not clear about the use of gnomAD prior to enrichment analysis

If SNVs that authors identify are true mosaic variants then they can't be common variants by definition. The chance that they find mosaic SNV that will match to the common variant in the population is extremely small (~0.1%). If their call set has significant overlap with gnomAD then the call set is of low quality, if the overlap is small then there is not need to prescreen variants against gnomAD.

Reviewer #3 (Remarks to the Author):

The authors have reasonably addressed my concerns. This is a very interesting study that is important to the field.

Reviewer #1

Remarks to the Author:

The authors have been extraordinarily responsive to my multiple questions, and have performed a series of additional control experiments to address my questions, all of which support their model. Furthermore, they expanded the data to include consideration of mutational signatures, clonal hematopoiesis and recent publications in the field of mosaicism in Alzheimer's to make the paper stronger.

Response:

We appreciate R1 for the approval of our previous responses and related positive comments on our revised manuscript.

Reviewer #2

Remarks to the Author:

I just have two minor comments

Response:

We appreciate R2 for the approval of our previous responses toward major questions. We addressed remaining minor concerns below.

MINOR QUESTIONS

Q1. Manual curation and BLAT results

I don't feel that the SNV chrX:136,877,960 C>T displayed in Figure R4b demonstrates the point. The panel displays two supporting reads for T in WES in liver and 6 supporting reads in WGS for brain. Why MuTect even called it? Isn't is suspicious? The SNV and indel next to it are always co-occurring and both of them have rsIDs (rs76812369 and rs755459969), which suggests that they may be germline. In this regard I think the authors should check and report how many of their putative mosaic variants are found in gnomAD.

Response:

We apologize for confusing description of the SNV (SCZ_1002; RBMX chrX:136,877,960 C>T) displayed in **Fig. R4b** from previous point-by-point response. We used targeted amplicon sequencing data of the SNV and 53 other variants called from raw MuTect call from 11 Schizophrenia cases to calibrate EBscore and set other post-filter conditions (e.g., cut-off value for BLAT score, necessity of visual inspection through IGV viewer). As R2 already noticed from the **Fig. R4b** which confirmed as "false call" from targeted amplicon sequencing, MuTect caller does not perform good enough for accurately detecting low-level somatic mutations from deep sequencing data due to the high false positive rate^{1,2}. Thereby, we adopted additional post-filter criteria (e.g., EBscore, Position of supporting reads, BLAT score, Visual inspection) to properly filter out those false calls from raw MuTect call before performing quantitative and qualitative assessment. By doing so, we could rule out possible germline variants and found 84.1~90.2% of post-filtered brain and blood somatic mutations in AD and non-AD cases are novel variants which are not reported in gnomAD (r2.0.2) exome databases (**Fig. RR1**). Even among those gnomAD-reported variants, most of them are ultra-rare variants (gnomAD MAF <0.01%): non-AD Brain (19/22), AD Brain (61/66), non-AD Blood (37/38), AD Blood (213/243).

a

Fig. RR1. The number of somatic mutations with gnomAD annotation in brain and blood tissues of AD and non-AD individuals.

a, Novel variants which are not reported in the all exome data-set from gnomAD (r2.0.2) are denoted as blue, ultra-rare variants with MAF <0.01% are denoted as orange, and rare variants with MAF <1% are denoted as red.

Q2. Operating by BLAT (or BLAST as the authors write in the caption to figure R4b) score values gives a reader no sense of how reasonable applied cut off are. The author should provide rough correspondence of cut off value to % of sequence identity.

Response:

Firstly, we apologize R2 for typos (c.f. “BLAST” should be edited to “BLAT” as mentioned elsewhere in revised manuscript) in the caption of **Fig. R4b** from previous point-by-point responses. We agree with R2 that BLAT score *per se* can be obscure to readers. The BLAT, BLAST-like alignment tool, account for both aligned length and sequence similarity when scoring alignment result of a query sequence³. The higher BLAT score mapped in one position, the more likely query sequence is uniquely located in that site of the genome. In case of 101-bp short sequencing reads (e.g., deep WES or TASEq) from Illumina sequencer, IGV-based BLAT score can be ranged from 0 to 1000 depending on the number of mismatch and gaps in the alignment and sequence identity between a query and the reference^{4,5}. Therefore, we preferred to use BLAT score by counting the number of mismatch and giving penalty on size of gaps, for detecting patterns of possible false calls, rather than simply using sequence identity (%).

The highest BLAT score of supporting reads from true calls validated from targeted amplicon sequencing, as seen in **Figs. RR2b and RR3b**, are >950 and such reads are appropriately aligned to in the vicinity of region-of-interest (see also **Supplementary Table 3**). In addition, the second highest BLAT score of supporting reads are <900, implying that there is no other location in genome that these reads can be mapped better. Meanwhile, the supporting reads from some of false calls proven

from targeted amplicon sequencing, as seen in **Figs. RR4b and RR5b**, the second highest BLAT score of supporting reads are >900 meaning that another location in the genome have similar sequence entity.

As R2 suggested, we could also rephrase our post-filter criteria of BLAT score from “the average second highest BLAT score should be <900” to “(assuming alignment of 101-bp long supporting reads have no gaps [spanning 101-bp]) each read has no more than one site with $\geq 95\%$ of sequence identity (%) to the reference genome”.

b

Read#	BLAT	span	Seq. Identity (%)	match	mis-match	query	chr	start	end	strand
1	980	101	99.1	100	1	ATAGAGGGCATTGCAGAAGCCCTCAATGGAGAGGCCACGGCCATAGTACTTGTTCCTGCAGAGGTATGCCCTGTGTCCAGCTGGTACACDCTGAAMCCCA	chr3	49727555	49727666	-
	257	28	96.5	27	1		chr17	72045118	72045146	-
2	980	101	99.1	100	1	TCAGGATAGGCTCAACAGGTCACGTCCAGGTCACGGCCATTGTGCAGATATTGATAGAGGGCATTGCAGAAGCCCTCAATGGAGAGGCCACGGCCATAG	chr3	49727510	49727611	-
	198	22	95.5	21	1		chr18	41934559	41934581	-
3	980	101	99.1	100	1	GTCACGGCCATTGTGCAGATATTGATAGAGGGCATTGCAGAAGCCCTCAATGGAGAGGCCACGGCCATAGTACTTGTTCCTGCAGAGGTATGCCCTGTGT	chr3	49727541	49727642	-
	267	30	100	28	0		chr17	72045116	72045146	-
4	980	101	99.1	100	1	AGGGCCCCAGTTTGTCCAGGATAGGCTCAACAGGTCACGTCCAGGTCACGGCCATTGTGCAGATATTGATAGAGGGCATTGCAGAAGCCCTCAATGGA	chr3	49727494	49727595	+
5	980	101	99.1	100	1	TCAGGATAGGCTCAACAGGTCACGTCCAGGTCACGGCCATTGTGCAGATATTGATAGAGGGCATTGCAGAAGCCCTCAATGGAGAGGCCACGGCCATAG	chr3	49727510	49727611	-
	198	22	95.5	21	1		chr18	41934559	41934581	-
6	980	101	99.1	100	1	ATTGCAGAAGCCCTCAATGGAGAGGCCACGGCCATAGTACTTGTTCCTGCAGAGGTATGCCCTGTGTCCAGCTGGTACACDCTGAAMCCCAAGGAGGAGA	chr3	49727574	49727675	-
	257	28	96.5	27	1		chr17	72045118	72045146	-
7	980	101	99.1	100	1	GCATTGCAGAAGCCCTCAATGGAGAGGCCACGGCCATAGTACTTGTTCCTGCAGAGGTATGCCCTGTGTCCAGCTGGTACACDCTGAAMCCCAAGGAGGCA	chr3	49727572	49727673	-
	257	28	96.5	27	1		chr17	72045118	72045146	-
8	980	101	99.1	100	1	CAGCTTTCAGGGCCCGCAGTTTGTCCAGGATAGGCTCAACAGGTCACGTCCAGGTCACGGCCATTGTGCAGATATTGATAGAGGGCATTGCAGAAGCCCT	chr3	49727486	49727587	+
9	980	101	99.1	100	1	CAGCACAGCTTTCAGGGCCCGCAGTTTGTCCAGGATAGGCTCAACAGGTCACGTCCAGGTCACGGCCATTGTGCAGATATTGATAGAGGGCATTGCAGA	chr3	49727481	49727582	+
10	980	101	99.1	100	1	GCAGTCCAGGGCCATTGTGCAGATATTGATAGAGGGCATTGCAGAAGCCCTCAATGGAGAGGCCACGGCCATAGTACTTGTTCCTGCAGAGGTATGCCCT	chr3	49727537	49727638	+
	227	23	100	23	0		chr17	72045123	72045146	-

Fig. RR2. Example of IGV browser and UCSC genome browser images of a true brain somatic mutation in *IP6K1* from Test dataset.

a, IGV browser image of true call show one mismatched base pair in each read from brain exome unless homozygous/heterozygous SNPs are located within 101-bp. **b**, The two highest BLAT scores of each supporting read with aligned length (span) and its sequence identity to reference genome (GRCh38). **c**, UCSC browser image of the reference genome-aligned supporting reads from brain sample where the BLAT score is the highest. **d**, UCSC browser image of the reference genome-aligned supporting reads from brain sample where the BLAT score is the second highest. Unlike reads shown in panel **c**, reads show gaps or partially aligned on the genome. Red vertical lines in panel **c** show query sequence and the reference genome have different bases at regarding position.

b

Read#	BLAT	Span	Seq. identity (%)	match	mis-match	query	chr	start	end	strand
1	980	101	99.1	100	1	GATGGTGGCGGGCGGCTGTGACACGCTTGACTCTGACTCTCATCTCGTCTGGTCCGCGACCTGATTTTCAGCAAGGAGGTGAGTCTCCCTTCCTCCCTCATCC	chrX	3311158	3311259	+
	861	98	99.8	93	5		chrY	11985845	11985943	-
2	980	101	99.1	100	1	GCAAGGAGGTTGTATCTCGCTTCCTCCATCCGACCTGGTAAAGTAGAGTGTCCGCAATTGTTGAAAGACGACATAGGGCTTGGTCCGCTCCACCGCTGTC	chrX	3311225	3311326	+
	861	101	93.1	54	7		chrY	11985778	11985879	-
3	980	101	99.1	100	1	GACTGTGACTCTCATCTCGTCTGGTCCGCGACCTGATTTTCAGCAAGGAGGTGAGTCTCCCTTCCTCCATCCGACCTGGTAAAGTAGAGTGTCCCAT	chrX	3311185	3311286	+
	881	101	94.1	95	6		chrY	11985918	11985919	-
4	980	101	99.1	100	1	GGTCTTCCGCGACCTGATTTTCAGCAAGGAGGTGAGTCTCGCTTCCTCCGCTCATCCGACCTGGTAAAGTAGAGTGTCCCATGTTGAAGACGACATAG	chrX	3311202	3311303	+
	861	101	93.1	94	7		chrY	11985801	11985902	-
5	980	101	99.1	100	1	GGTGGCGGGGGCGTGCACGACCTTGACTCTGACTCTCATCTCGTCTGGTCCGCGACCTGATTTTCAGCAAGGAGGTGAGTCTCCCTTCCTCCATCCGCA	chrX	3311161	3311262	+
	801	89	95.6	85	4		chrY	11985842	11985931	-
6	980	101	99.1	100	1	GGGCGTGTGACGACCTTGACTCTGACTCTCATCTCGTCTGGTCCGCGACCTGATTTTCAGCAAGGAGGTGAGTCTCCCTTCCTCCATCCGCACTTGGTT	chrX	3311168	3311269	+
	841	93	95.7	89	4		chrY	11985838	11985931	-
7	980	101	99.1	100	1	GTCACGACGTTGACTCTGACTCTCATCTCGTCTGGTCCGCGACCTGATTTTCAGCAAGGAGGTGAGTCTCCCTTCCTCCATCCGCACTTGGTTAAAGTA	chrX	3311174	3311275	+
	831	92	95.7	88	4		chrY	11985838	11985930	-

Fig. RR3. Example of IGV browser and UCSC genome browser images of a true brain somatic mutation in *MXRA5* from Test dataset.

a, IGV browser image of true call show one mismatched base pair in each read from brain exome unless homozygous/heterozygous SNPs are located within 101-bp. **b**, The two highest BLAT scores of each supporting read with aligned length (span) and its sequence identity to reference genome (GRCh38). **c**, UCSC browser image of the reference genome-aligned supporting reads from brain sample where the BLAT score is the highest. **d**, UCSC browser image of the reference genome-aligned supporting reads from brain sample where the BLAT score is the second highest. Unlike reads shown in panel **c**, reads show gaps, mismatches or partially aligned on the genome. Red vertical lines in panel **c** show query sequence and the reference genome have different bases at regarding position.

a

Subject ID: SCZ1002
 Gene: RBMX
 Mutation: chrX:136877960 C>T
 EBScore: 0.118
 Top1 BLAT (avg.): 930-950 (942)
 Top2 BLAT (avg.): 900-930 (925)

deep WESeq
 Liver
 PFC
 TASEq
 Liver
 PFC

b

Read#	BLAT	Span	Seq. Identity (%)	mismatch	mis-match	query	chr	start	end	strand
1	930	99	94.0	97	2	ATATCCACCGTCATCCATGTGCTCCCGGTGAGGGAGGTCCTGGTTCCTCCACTTCTCTCTCCACCTCTAGACCTCTGGAAAGGGCCCTACT	chr9	30689407	30689506	-
	900	100	94.0	96	3		chrX	85996733	85996893	+
2	930	99	94.0	97	2	ATCCATGTGTCCTCCCGGTGAGGGAGGTCCTGGTTCCTCCACTTCTCTCTCCACCTCTAGACCTCTGGAAAGGGCCCTACTCTCTGGAGGTGG	chr9	30689356	30689434	-
	930	99	94.0	97	2		chrX	136877913	136878011	-
3	940	222	98.0	98	1	CTTCTCTCTCTCCACCTCTAGACCTCTGGAAAGGGCCCTACTCTCTGGAGGTGGAGGGCCCTCCACCTCTCCACTTTCAATGATGTTGGTGGCT	chr9	30689362	30689574	-
	930	99	94.0	97	2		chr4	109347115	109347214	+
4	950	99	98.0	98	1	TGTCTCCCGGTGAGGGAGGTCCTGGTTCCTCCACTTCTCTCTCCACCTCTAGACCTCTGGAAAGGGCCCTACTCTCTGGAGGTGGAGGGCT	chrX	136877919	136878018	+
	930	99	94.0	97	2		chr9	30689388	30689487	-
5	950	99	98.0	98	1	AGTCCCGCTGGTTCCTCCACTTCTCTCTCCACCTCTAGACCTCTGGAAAGGGCCCTACTCTCTGGAGGTGGAGGGCCCTCCACCTCTCCACTTTC	chr9	136877936	136878035	+
	930	99	94.0	97	2		chr9	30689371	30689470	-
6	950	99	98.0	98	1	CTTCTCTCTCTCCACCTCTAGACCTCTGGAAAGGGCCCTACTCTCTGGAGGTGGAGGGCCCTCCACCTCTCCACTTTCAATGATGTTGGTGGCTTGT	chrX	136877969	136878067	-
	930	99	94.0	97	2		chr9	30689349	30689448	-

c

d

Fig. RR4. Example of IGV browser and UCSC genome browser images of a false brain somatic mutation in *RBMX* from Test dataset.

a, IGV browser image of true call show one mismatched base pair in each read from brain exome unless homozygous/heterozygous SNPs are located within 101-bp. **b**, The two highest BLAT scores of each supporting read with aligned length (span) and its sequence identity to reference genome (GRCh38). **c**, UCSC browser image of the reference genome-aligned supporting reads from brain sample where the BLAT score is the highest. **d**, UCSC browser image of the reference genome-aligned supporting reads from brain sample where the BLAT score is the second highest. Unlike reads shown in **Figs. RR2d and RR3d**, these reads show comparable sequence identity to other site of the genome as well. Red vertical lines in panel **c** show query sequence and the reference genome

have different bases at regarding position. Orange vertical lines in panels **c**, **d** show the query sequence has an insertion (or genome has a deletion/alignment gap) at this point. Purple vertical lines implying that query sequence extends beyond the end of the alignment.

b

Read#	BLAT	span	Seq. Identity (%)	match	mis-match	query	chr	start	end	strand
1	980	101	99.1	100	1	GCTCAGTCATGTCCTTTGATTTTATGCTGTGGAGAGAAATGGCCGAGTTTCATCACCCCTTCTCTAGCCAGCCTCCTCCGCGTCCAGACACGGGAGA	chr7	75426380	75426481	-
	980	101	99.1	100	1		chr7	72938561	72938662	+
2	1000	101	100	101	0	AGAGACTTTGTTCATGTTGAGGGTCAGATGCTAAGTTATCAGCAGGCTCAGTCATGTCCTCTTATTTTATGCTGTGGAGAGAAATGGCCGAGTTTCATC	chr7	75426424	75426525	-
	1000	101	100	101	0		chr7	73295434	73295535	+
3	1000	101	100	101	0	GGTCAGATGCTAAGTTATCAGCAGGCTCAGTCATGTCCTCTTATTTTATGCTGTGGAGAGAAATGGCCGAGTTTCATCACCCCTTCTCTAGCCAGCCT	chr7	75426404	75426505	-
	1000	101	100	101	0		chr7	73293444	73293545	+

Fig. RR5. Example of IGV browser and UCSC genome browser images of a false brain somatic mutation in *POM121* from Test dataset.

a, IGV browser image of true call show one mismatched base pair in each read from brain exome unless homozygous/heterozygous SNPs are located within 101-bp. **b**, The two highest BLAT scores of each supporting read with aligned length (span) and its sequence identity to reference genome (GRCh38). **c**, UCSC browser image of the reference genome-aligned supporting reads from brain sample where the BLAT score is the highest. **d**, UCSC browser image of the reference genome-aligned supporting reads from brain sample where the BLAT score is the second highest. Unlike reads in **Figs. RR2d and RR3d**, these reads show comparable sequence identity to other site of the genome as well. Red vertical lines in panel **c** show query sequence and the reference genome have different bases at regarding position.

Q3. *I'm still not clear about the use of gnomAD prior to enrichment analysis*

If SNVs that authors identify are true mosaic variants then they can't be common variants by definition. The chance that they find mosaic SNV that will match to the common variant in the population is extremely small (~0.1%). If their call set has significant overlap with gnomAD then the call set is of low quality, if the overlap is small then there is not need to prescreen variants against gnomAD.

Response:

We apologize for unclear description in reasoning of prescreening common variants from post-filtered calls when performing enrichment test. As we described in **Fig. RR1**, the most of post-filtered calls are already novel or rare variants and prescreening with gnomAD <0.01% criteria only filter out 0.28~2.17% of variants from brain and blood SNVs from AD and non-AD cases. Nevertheless, we performed enrichment test without filtering out such common variants to confirm the significance of our previous findings. As seen in **Fig. RR6b**, the gene-set enrichment result from Enrichr still show significance in PI3K-AKT (Top1, adj. $p < 0.0001$; 15/341 overlap), MAPK (Top3, $p = 0.0009$; 11/255), and AMPK (Top6, $p = 0.0039$; 7/124) pathways only in AD brain. Moreover, we showed that gene-length bias did not affect the enrichment test result when using such gene-set as well (**Fig. RR6d**).

Fig. RR6. Enrichment test results of genes with pathogenic mutations.

Enrichment test was performed with two independent methods using KEGG (v.2016) pathways. Enrichment test result with exclusion of common variants (gnomAD>0.01%) showed three significant hyperphosphorylation of Tau-related pathways from both **a**, Enrichr and **c**, gene-length adjusted DNENRICH method. Putatively pathogenic gene-set with inclusion of common variants still showed significance in the three pathways from both **b**, Enrichr and **d**, DNENRICH methods. Gene-length adjusted enrichment test was performed with DNENRICH along with 100,000 permutations. Vertical bar represents threshold for Benjamini-Hochberg adjusted P value.

Reviewer #3**Remarks to the Author:**

The authors have reasonably addressed my concerns. This is a very interesting study that is important to the field.

Response:

We thank R3 for the approval of our previous responses and related positive comments on our revised manuscript.

References

1. Kim, J. *et al.* The use of technical replication for detection of low-level somatic mutations in next-generation sequencing. *Nature Communications* **10**, 1047 (2019).
2. Krøigård, A.B., Thomassen, M., Lænkholm, A.-V., Kruse, T.A. & Larsen, M.J. Evaluation of Nine Somatic Variant Callers for Detection of Somatic Mutations in Exome and Targeted Deep Sequencing Data. *PloS one* **11**, e0151664-e0151664 (2016).
3. Kent, W.J. BLAT—The BLAST-Like Alignment Tool. *Genome Research* **12**, 656-664 (2002).
4. Robinson, J.T., Thorvaldsdóttir, H., Wenger, A.M., Zehir, A. & Mesirov, J.P. Variant Review with the Integrative Genomics Viewer. *Cancer Research* **77**, e31-e34 (2017).
5. Nishioka, M. *et al.* Identification of somatic mutations in postmortem human brains by whole genome sequencing and their implications for psychiatric disorders. *Psychiatry and Clinical Neurosciences* **72**, 280-294 (2018).

REVIEWERS' COMMENTS:

Reviewer #2 (Remarks to the Author):

The authors' response is reasonable.

Reviewer #2

Remarks to the Author:

The authors' response is reasonable.

Response:

We appreciate R2 for the approval of our previous responses on our revised manuscript.